# Mechanisms of distributed working memory in a large-scale network of macaque neocortex

**Jorge F Mejías[1], Xiao-Jing Wang[2]***

[1]Swammerdam Institute for Life Sciences, University of Amsterdam, Amsterdam, Netherlands; [2]Center for Neural Science, New York University, New York, United States

**Abstract:** Neural activity underlying working memory is not a local phenomenon but distributed across multiple brain regions. To elucidate the circuit mechanism of such distributed activity, we developed an anatomically constrained computational model of large-scale macaque cortex. We found that mnemonic internal states may emerge from inter-areal reverberation, even in a regime where none of the isolated areas is capable of generating self-sustained activity. The mnemonic activity pattern along the cortical hierarchy indicates a transition in space, separating areas engaged in working memory and those which do not. A host of spatially distinct attractor states is found, potentially subserving various internal processes. The model yields testable predictions, including the idea of counterstream inhibitory bias, the role of prefrontal areas in controlling distributed attractors, and the resilience of distributed activity to lesions or inactivation. This work provides a theoretical framework for identifying large-scale brain mechanisms and computational principles of distributed cognitive processes.

***For correspondence:**
xjwang@nyu.edu

**Competing interest:** The authors declare that no competing interests exist.

## Editor's evaluation

The final revision of the manuscript addressed the remaining issues raised by the reviewers. They felt that the paper is an important contribution to the field, providing new and testable insights into the interaction between cortical areas during the memory delay and that the work is likely to become "an influential reference for future modeling efforts" and deserves publication in *eLife*.

## Introduction

With the advances of brain connectomics and physiological recording technologies like Neuropixels (*Jun et al., 2017*; *Stringer et al., 2019*), an increasingly important challenge in Neuroscience is to investigate biological mechanisms and computational principles of cognitive functions that engage many interacting brain regions. Here, the goal is no longer to identify one local parcellated brain region that contributes to or is crucial for a particular function, but how a large-scale brain system with many interacting parts underlie behavior. Currently, there is a dearth of theoretical ideas and established models for understanding distributed brain dynamics and function.

A basic cognitive function recently shown to involve multiple brain areas is working memory, the brain's ability to retain and manipulate information in the absence of external inputs. Working memory has been traditionally associated with mnemonic delay neural firing in localized brain areas, such as those in the frontal cortex (*Funahashi et al., 1989*; *Fuster, 1973*; *Goldman-Rakic, 1995*; *Inagaki et al., 2019*; *Kopec et al., 2015*; *Romo et al., 1999*) and computational models uncovered the involvement of local recurrent connections and NMDA receptors in the encoding of memory items in

selective neural assemblies (*Amit and Brunel, 1997*; *Brunel and Wang, 2001*; *Compte et al., 2000*; *Wang et al., 2013*; *Wang, 1999*).

In spite of the prevalence of prefrontal cortex as a 'hub' for working memory maintenance, self-sustained neural activity during working memory has been found in multiple brain regions; and often such highly engaged areas appear in coactivation (*Christophel et al., 2017*; *Guo et al., 2017*; *Leavitt et al., 2017*; *Sreenivasan and D'Esposito, 2019*). Previous modeling efforts have been limited to exploring either the emergence of sustained activity in local circuits, or in two interacting areas at most (*Edin et al., 2009*; *Guo et al., 2017*; *Murray et al., 2017b*). It is presently not known what biophysical mechanisms could underlie a distributed form of memory-related sustained activity in a large-scale cortex. The observation that mnemonic activity is commonly found in the prefrontal cortex (PFC) does not prove that it is produced locally rather than resulting from multi-regional interactions; conversely, a distributed activity pattern could in principle be a manifestation of sustained inputs broadcasted by a local source area that can produce self-sustained activity in isolation. Therefore, understanding the distributed nature of cognitive functions such as working memory is challenging and requires of both novel theoretical ideas and multi-area recordings (*Edin et al., 2009*; *Guo et al., 2017*).

In this study, we tackle this challenge by building and analyzing an anatomically constrained computational model of the cortical network of macaque monkey, and investigate a novel scenario in which long-range cortical interactions support distributed activity patterns during working memory. The anatomical data is used to constrain the model at the level of long-range connections but also at the level of local circuit connectivity. In particular, the model incorporated differences between individual cortical areas, by virtue of macroscopic gradients of local circuit properties in the large-scale network. The emerging distributed patterns of sustained activity involve many areas across the cortex. They engage temporal, frontal and parietal areas but not early sensory areas, in agreement with a recent meta-analysis of delay period activity in macaque cortex (*Leavitt et al., 2017*). Sustained firing rates of cortical areas across the hierarchy display a gap, indicative of the existence of a transition akin to a bifurcation in cortical space that does not require tuning of bifurcation parameters. Furthermore, the distributed patterns emerge even when individual areas are unable to maintain stable representations, or when other mechanisms such as activity-silent memory traces are considered. Our model predicts that distributed WM patterns (i) require the existence of a certain level of inhibition in long-range feedback projections, (ii) can be controlled or inactivated from a small group of areas at the top of the cortical hierarchy, and (iii) increase the robustness of the network to distractors and simulated inactivation of areas. The concept departs from the classical view of working memory based on local attractors, and sheds new light into recent evidence on distributed activity during cognitive functions.

## Results

Our computational model includes 30 areas distributed across all four neocortical lobes (*Figure 1A*; see Materials and methods for further details). The inter-areal connectivity is based on quantitative connectomic data from tract-tracing studies of the macaque monkey (*Markov et al., 2013*; *Markov et al., 2014a*; *Markov et al., 2014b*; *Figure 1—figure supplement 1*). For simplicity, each of the cortical areas is modeled as a neural circuit which contains two excitatory populations (selective to sensory stimuli A and B, respectively) and one inhibitory population (*Wang, 2002*; *Wang, 1999*; *Figure 1B*). In addition, the model assumes a macroscopic gradient of outgoing and recurrent synaptic excitation (*Chaudhuri et al., 2015*; *Joglekar et al., 2018*; *Wang, 2020*), so that the level of synaptic strength is specific of each area (*Figure 1—figure supplement 2*). This gradient is introduced by considering that the number of apical dendritic spines, loci of excitatory synapses, per pyramidal cells increases (*Elston, 2007*) along the cortical hierarchy as defined by anatomical studies (*Felleman and Van Essen, 1991*; *Markov et al., 2014a*; *Figure 1C*). The gradient of area-specific connection strength was applied to both local recurrent and long-range excitatory outgoing projections. In particular, we denote the maximal strength of local and long-range excitation for the area at the top of the cortical hierarchy by $J_{max}$, which is an important parameter of the model (see below). To allow for the propagation of activity from sensory to association areas, we assumed that inter-areal long-distance outgoing projections target more strongly excitatory neurons for feedforward pathways and inhibitory neurons for feedback pathways, in a graded fashion (*Mejías et al., 2016*; *Figure 1B*). We shall refer to the gradual preferential targeting onto inhibitory neurons by top-down projections as the 'counterstream inhibitory bias' hypothesis. We assume that the bias of top-down

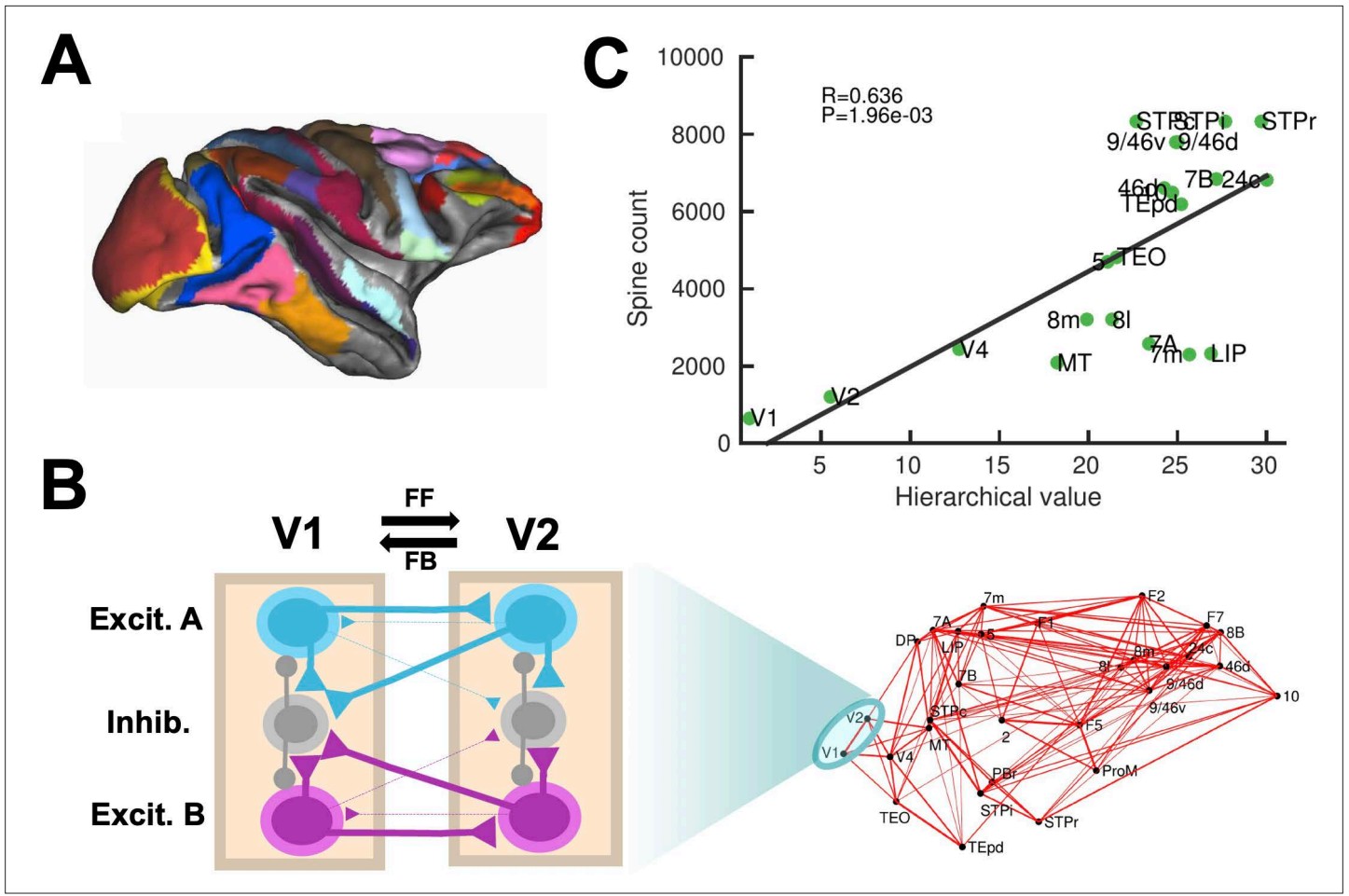

**Figure 1.** Scheme and anatomical basis of the multi-regional macaque neocortex model. (**A**) Lateral view of the macaque cortical surface with modelled areas in color. (**B**) In the model, inter-areal connections are calibrated by mesoscopic connectomic data (*Markov et al., 2013*), each parcellated area is modeled by a population firing rate description with two selective excitatory neural pools and an inhibitory neural pool (*Wong and Wang, 2006*). Recurrent excitation within each selective pool is not shown for the sake of clarity of the figure. (**C**) Correlation between spine count data (*Elston, 2007*) and anatomical hierarchy as defined by layer-dependent connections (*Markov et al., 2014a*).

The online version of this article includes the following figure supplement(s) for figure 1:

**Figure supplement 1.** Anatomical connectivity data of the macaque cortex.

**Figure supplement 2.** Spine count data used to constrain connectivity strength.

projections towards inhibitory neurons is proportional to the fraction of infragranular projections (see Materials and methods). It is worth noting that exploration of such new hypotheses would have not been possible without a quantitative definition of the cortical hierarchy and biologically realistic circuit modeling. The results provided by this anatomically constrained model, while leading to concrete experimental predictions for macaques, are also robust to small alterations of parameter values and connectivity structure, suggesting the validity of our conclusions in other conditions or animal species.

## Distributed WM is sustained by long-range cortical loops

In local circuit models of working memory (WM)(*Compte et al., 2000*; *Wang, 1999*), areas high in the cortical hierarchy make use of sufficiently strong synaptic connections (notably involving NMDA receptors *Wang et al., 2013*; *Wang, 1999*) to generate self-sustained delay activity. Specifically, the strength of local synaptic reverberation must exceed a threshold level (in our model, the local coupling parameter $J_S$ must be larger than a critical value of 0.4655), for an isolated local area to produce stimulus-selective mnemonic activity states that coexist with a resting state of spontaneous activity (operating in a multistable regime rather than in a monostable regime, see *Figure 2A*). However,

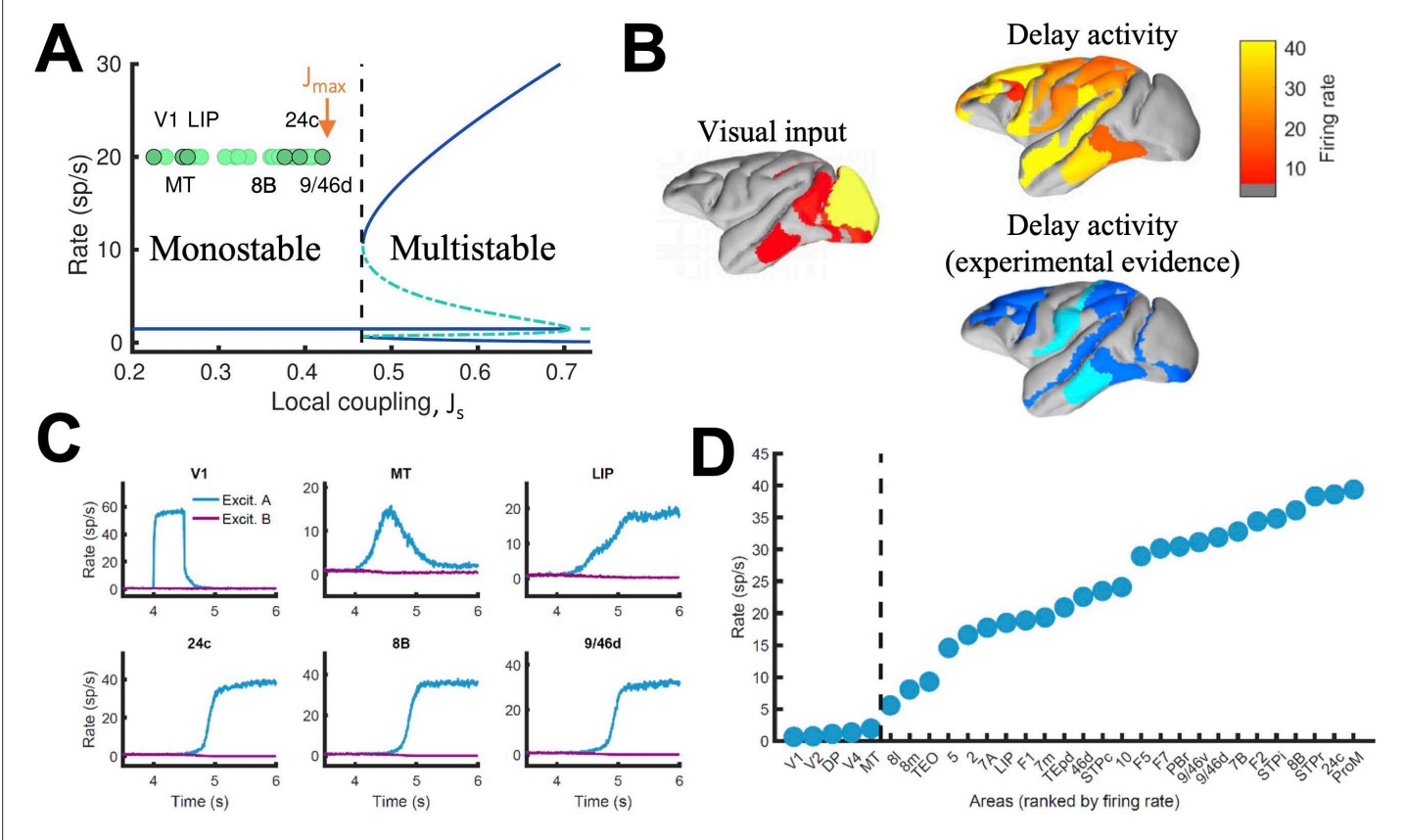

**Figure 2.** Distributed WM sustained via long-range loops in cortical networks. (**A**) Bifurcation diagram for an isolated area. Green circles denote the position of each area, with all of them in the monostable regime when isolated. (**B**) Spatial activity map during visual stimulation (left) and delay period (upper right). For comparison purposes, bottom right map summarizes the experimental evidence of WM-related delay activity across multiple studies (*Leavitt et al., 2017*), dark blue corresponds to strong evidence and light blue to moderate evidence. (**C**) Activity of selected cortical areas during the WM task, with a selective visual input of 500ms duration. (**D**) Firing rate for all areas during the delay period, ranked by firing rate.

The online version of this article includes the following figure supplement(s) for figure 2:

**Figure supplement 1.** Behavior of all areas in the network during the visual WM task.

**Figure supplement 2.** Behavior of all areas in the network during the somatosensory WM task.

**Figure supplement 3.** A simple gating mechanism controls the participation of areas in distributed WM.

**Figure supplement 4.** Firing rates for selected areas during a visual WM task with a short (50ms) stimulus duration.

**Figure supplement 5.** Firing rate of ranked cortical areas reveals a robust transition in space with different ranking systems.

there is presently no conclusive experimental demonstration that an isolated cortical area like dorsolateral prefrontal cortex (dlPFC) is indeed capable of generating mnemonic sustained activity. Indeed, recent evidence suggest that thalamocortical support might be needed to achieve sustained activity in dlPFC (*Guo et al., 2017*). In this study, we first examined the scenario in which all areas, including dlPFC (9/46d) at the top of the hierarchy, have $J_S$ values below the critical value for multistability (so $J_S \leq J_{max} < 0.4655$) and are connected via excitatory long-range projections of global coupling strength G (we set $J_{max} = 0.42$ and G = 0.48 unless specified otherwise)(*Figure 2A*). In this case, any observed sustained activity pattern must result from inter-areal connection loops. In a model simulation of a visual delayed response task, a transient visual input excites a selective neural pool in the primary visual cortex (V1), which yielded activation of other visual areas such as MT during stimulus presentation (*Figure 2B*, upper left). After stimulus withdrawal, neural activity persists in multiple areas across frontal, temporal and parietal lobes (*Figure 2B*, top right). The resulting activation pattern shows a substantial agreement with a large body of data, from decades of monkey neurophysiological experiments, reviewed in recent meta-analyses (*Christophel et al., 2017*; *Leavitt et al., 2017*), regarding which areas display WM-related activity during delay period of WM tasks

(*Figure 2B*, bottom right). The activation pattern from the model was stimulus specific, so only the neural pool selective to the presented stimulus in each cortical area displayed elevated sustained activity (*Figure 2C*; *Figure 2—figure supplement 1*). We observed cross-area variations of neural dynamics: while areas like 9/46d displayed a sharp binary jump of activity, areas like LIP exhibited a more gradual ramping activity. Such a population level, or neuron-averaged, ramping activity of LIP in our model would correspond to the trial-averaged temporal accumulation of information in decision-making (*Shadlen and Newsome, 2001*).

Given that selective mnemonic activity is also found in somatosensory WM tasks (*Romo et al., 1999*), we further test our model and simulate a simple somatosensory WM task by transiently and selectively stimulating a neural pool in primary somatosensory cortex. As in the case of visual stimulation, this leads to the emergence of a distributed sustained activity pattern of equal selectivity as the input (*Figure 2—figure supplement 2*), showing the validity of the distributed WM mechanism across different sensory modalities. At this stage, the model is however not able to predict different between attractors evoked by different sensory modalities. For this, we show that the introduction of a simple gating mechanism allows to study the involvement of certain areas in the processing of particular input modalities, further refining the model predictions (*Figure 2—figure supplement 3*). Likewise, our model has considered NMDA receptors as the only excitatory dynamics for simplicity. However, AMPA dynamics may also be important (*van Vugt et al., 2020*), and can be easily introduced leading

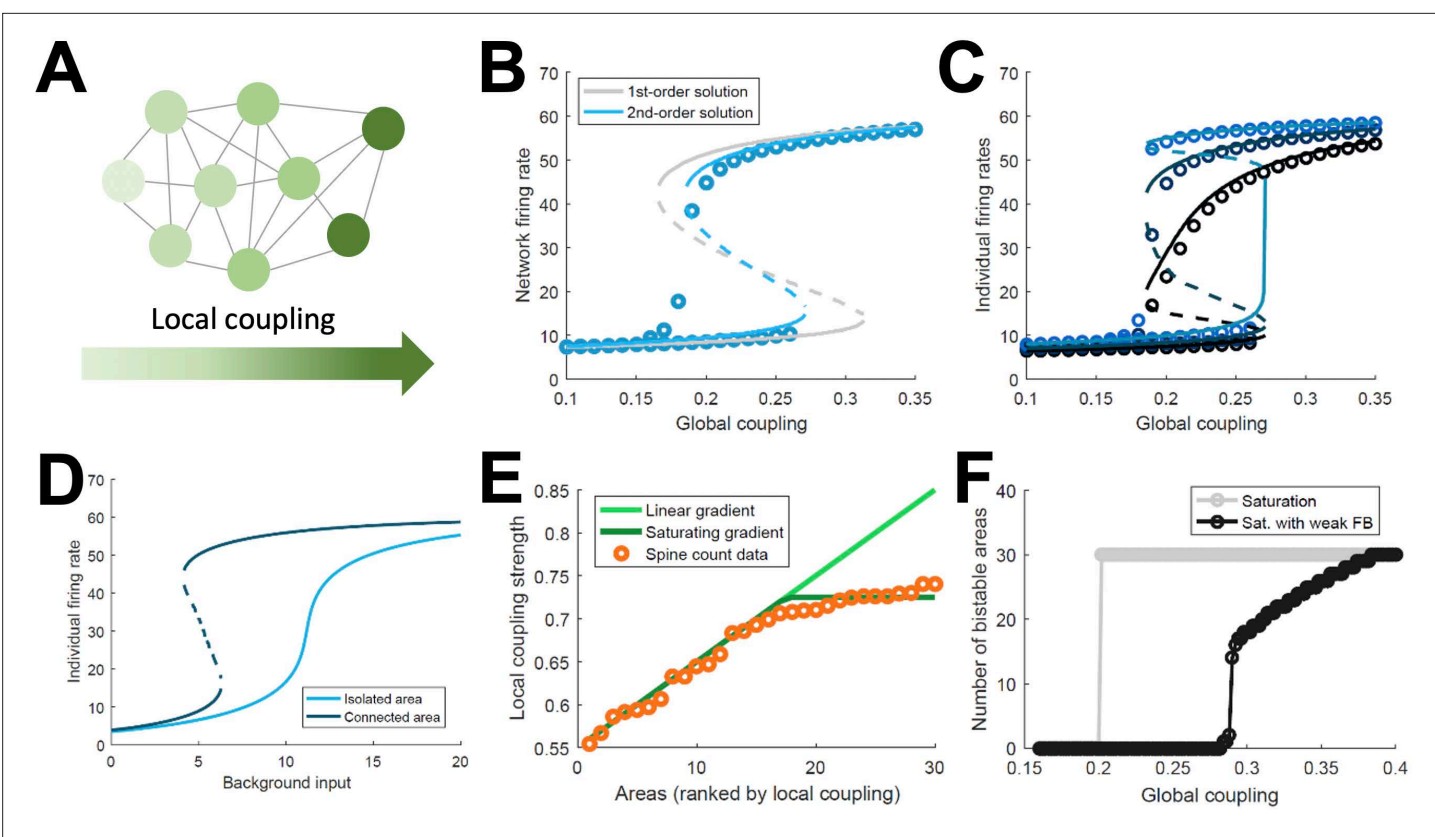

**Figure 3.** Simplified model of distributed WM. (**A**) Scheme of our simplified model: a fully connected network of N = 30 excitatory nodes with a gradient of local coupling strength. (**B**) Population-average firing rate as a function of the global coupling strength, according to numerical simulations (symbols) and a mean-field solution based on first-order (gray line) or second-order statistics (blue). (**C**) Firing rates of three example individual nodes (symbols denote simulations, lines denote second-order mean-field solutions). (**D**) Activity of an example node when isolated (light blue) or connected to the network (dark blue). (**E**) Two forms for the gradient of local coupling strength (lines) compared with the spine count data (symbols). (**F**) The number of bistable areas in the attractor (grey) is either zero or N when the gradient of local properties saturates as suggested by spine count data. When feedback projections become weaker, resembling inhibitory feedback, the network is able to display distributed WM patterns which involve only a limited number of areas (black).

The online version of this article includes the following figure supplement(s) for figure 3:

**Figure supplement 1.** Simplified network model for distributed, activity-silent memory traces.

to a good behavior of the model for shorter durations of the stimulus presentation (*Figure 2—figure supplement 4*).

When we plotted the firing rate of stimulus-selective sustained activity across 30 areas along the hierarchy, our results revealed a separation between the areas displaying sustained activity and those that did not (*Figure 2D*). This is a novel type of abrupt transition of behavior that takes place in space, rather than as a function of a network parameter like in *Figure 2A*. As a matter of fact, the relevant parameter here is the strength of synaptic excitation that varies across cortical space, in the form of a macroscopic gradient. The transition is robust in two respects. First, the separation between areas appears not only when areas are ranked according to their firing rates, but also when they follow their positions in the anatomical hierarchy or in the rank of spine count values (*Figure 2—figure supplement 5*). Second, it does not depend on any fine tuning of parameter values.

## Simplified model of distributed working memory

The above model, albeit a simplification of real brain circuits, includes several biologically realistic features, which makes it difficult to identify essential ingredients for the emergence of distributed WM. For this reason, we developed a minimal model consisting on a fully connected network of excitatory firing-rate nodes (*Figure 3A*, see Appendix 1). This simplified model will allow us to explore the minimal conditions for the emergence of distributed WM, in the same way that the full, biologically realistic model provided us with stronger support for the mechanism in realistic conditions. The network of the simplified model includes a linear gradient of local properties: areas at the beginning of such gradient have weak self-coupling, while areas at the end have strong self-coupling. As in the more elaborated model, self-excitation is too weak to generate bistability in any isolated nodes.

This simple model allows for a mean-field analytical solution for the network average firing rate R of the form: $R = \phi \left( \left( J\eta_0 + G \right) R + I \right)$, with $\phi$ being a sigmoidal function, $J\eta_0$ the average local coupling value across areas, $G$ the inter-areal connection strength, and $I$ a background input current (see Appendix 1 for a full derivation of a two-area example and a more complete N-area network). The factor $J\eta_0 + G$ determines whether the above equation has one stable solution (spontaneous firing) or two (spontaneous and sustained firing). As this factor includes both local and global components, the average network firing rate may be bistable even if local couplings are weak, as long as the inter-areal connections are strong enough. This mean-field solution, as well as a more precise second-order version, show a good agreement with numerical simulations and confirm the emergence of distributed activity in the system (*Figure 3B*). Simulations also show, around $G \sim 0.17$, the appearance of states in which only areas at the top of the gradient show bistability, indicated by low values of R. Once R is known, the mean-field solution also permits to predict the emergence of sustained activity in individual nodes (*Figure 3C*) and also observe how monostable isolated units become bistable when incorporated into the network (*Figure 3D*). Briefly, the existence of a clear bistability in R would induce, in the individual areas with stronger self-coupling, the emergence of bistability for these areas. For areas with weak self-coupling, either possible value of R is not enough to induce local bistability, and these areas remain in the monostable regime.

The simplified model demonstrates that distributed WM patterns, in which some (but not all) areas display bistability when connected to each other, may emerge on generic networks of excitatory units as long as (i) their long-range connections are strong enough and (ii) the network has a linear gradient of local couplings. When considering biological constraints, however, these two conditions might not be easy to meet. In particular, data on the area-specific number of spines per neuron seems to monotonically increase, but saturates instead of growing linearly (*Figure 3E*). Introducing this saturating gradient on the simplified model makes the nodes more homogeneous, and as a result the network is not able to display distributed WM patterns without indistinctively activating all nodes (*Figure 3F*, gray curve). This problem was solved when we assumed that feedforward projections (i.e. those going from lower to higher areas in the gradient) were slightly stronger while feedback projections were slightly weaker, which is consistent with the counterstream inhibitory bias hypothesis. Such assumption, needed for saturating gradients, allows to recover solutions in which only a subset of areas display bistability in the WM patterns (*Figure 3F*, black curve).

Simplified models can also be used to explore the plausibility of the distributed WM hypothesis in other scenarios besides delay activity –for example, for activity-silent memory traces (*Kamiński and Rutishauser, 2020*; *Mongillo et al., 2008*; *Stokes, 2015*). We modified the simplified model

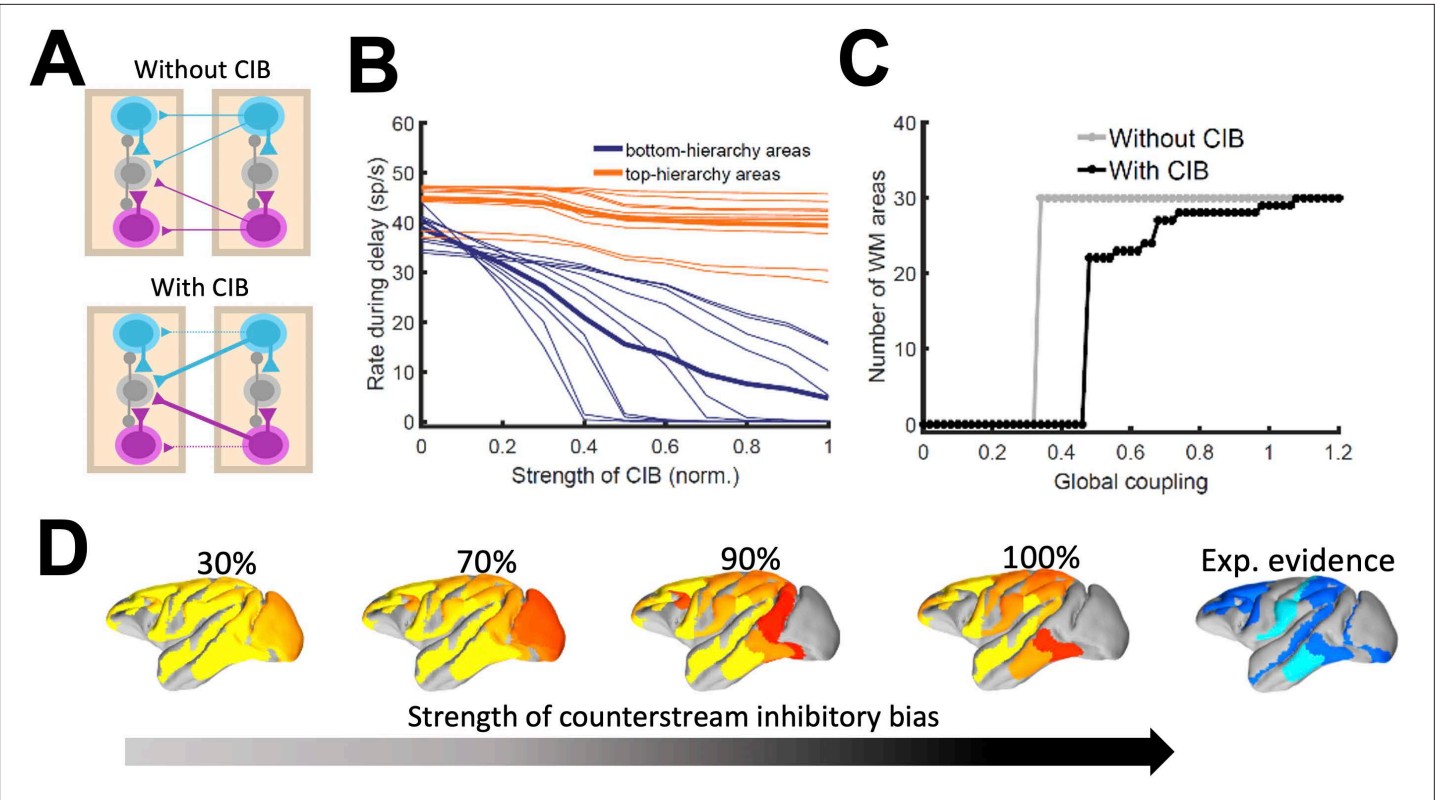

**Figure 4.** Effect of inhibitory feedback on distributed WM for the full model. (**A**) Scheme showing a circuit without (left) or with (right) the assumption of counterstream inhibitory bias, or CIB. (**B**) Firing rate of areas at the bottom and top of the hierarchy (10 areas each, thick lines denote averages) as a function of the CIB strength. (**C**) Number of areas showing sustained activity in the example distributed activity pattern vs global coupling strength without (grey) and with (black) CIB. (**D**) Activity maps as a function of the CIB strength. As in *Figure 2* B, bottom map denotes the experimental evidence for each area (dark blue denotes strong evidence, light blue denotes moderate evidence).

The online version of this article includes the following figure supplement(s) for figure 4:

**Figure supplement 1.** Corrections to localized regions; example of FEF areas.

**Figure supplement 2.** Effect of jitter on dendritic spine values and surrogate networks.

**Figure supplement 3.** A gradient of time scales emerges in the network.

introduced above by adding short-term synaptic facilitation (STF) to both local and long-range projections of the network (and decreasing the overall synaptic strength to allow the transient enhancements of STF to play a sufficient role; see Appendix 1). In such a model, a slowly decaying transient of synaptic efficacy, susceptible of reactivations along the delay period, is thought to preserve the information in an activity-silent manner. As in the delay activity model, isolated areas without STF are not able to sustain the information; similar results are obtained when STF is introduced. However, when long-range projections are considered in a network with STF, synaptic efficacies are sustained for long periods of time (as a result of the contribution of both local and long-range interactions), leading to areas with strong enough synapses to preserve the information via activity-silent memory traces (*Figure 3—figure supplement 1*).

### Impact of the counterstream inhibitory bias in the full model

As indicated by the simplified model, introducing differences between feedforward and feedback projections is a key ingredient to achieve realistic patterns of distributed WM in a data-constrained model. In the full, biologically more realistic model –which will be considered for the rest of the study –this asymmetry is introduced by considering a graded preferential targeting to inhibitory neurons by top-down projections (i.e. counterstream inhibitory bias, or CIB), which prevent indiscriminate sustained activation across all cortical areas (*Leavitt et al., 2017*; *Figure 4A*). We systematically varied the strength of the feedback projections targeting inhibitory population in our model, and computed

the firing rates of different areas during delay for these cases. We observed that, for strong enough CIB, the overall firing rate of early sensory areas is reduced, while the activity levels of areas high in the hierarchy is maintained at appropriate values (*Figure 4B*). This also allows distributed WM patterns to emerge for a wide range of the global coupling strength (*Figure 4C*).

The strength of the counterstream inhibitory bias has also an impact on the overall activity profiles across the brain network. *Figure 4D* shows activity maps for several CIB level, revealing that a moderate to strong inhibitory feedback agrees well with the experimental evidence.

It is also worth noting that exceptions to the CIB rule may exist in brain networks without compromising the stability of distributed WM attractors. For example, a more balanced targeting would allow for WM-related activity in primary visual areas, which still constitutes a point of controversy in the field (*Leavitt et al., 2017*). FEF areas 8 l and 8 m, on the other hand, are not able to sustain delay activity when receiving strong inhibitory feedback (especially from other frontal areas) and had to be excluded from this general rule, although such exception does not affect the results aside from local effects in FEF (Materials and Methods, *Figure 4—figure supplement 1*).

In *Figure 2* and also in the following sections, the strength of the counterstream inhibitory bias was considered proportional to the fraction of infragranular projections, as suggested by anatomical studies (*Markov et al., 2014b*) and following previous work (*Mejias et al., 2016*). This results in a very small bias for most of the projections, but enough to produce the desired effect (see Materials and methods for further details). In addition to supporting the emergence of distributed WM, CIB could explain observed top-down inhibitory control effects (*Tsushima et al., 2006*).

While the macroscopic gradient of excitability is an important property of the model, the particular values of excitatory strength assigned to each area are not relevant for the phenomenon (*Figure 4—figure supplement 2* A-D). Similar conclusions can be obtained when the anatomical structure of the cortical network is changed –for example, by randomly shuffling individual projection strength values (*Figure 4—figure supplement 2E-H*). However, in this case, the duration of the sustained activity for multiple areas may be affected. This suggests that salient statistical features of the structure embedded in the cortical network may play a role in the emergence of distributed activity patterns. The model also predicts the emergence of a hierarchy of time scales across cortical areas (*Figure 4—figure supplement 3*), in agreement with experimental findings (*Murray et al., 2014*) and supporting and improving previous computational descriptions (*Chaudhuri et al., 2015*).

## Long-range cortical loops support a large number of different distributed attractors

We realized that a large-scale circuit can potentially display a large number of distributed sustained activity patterns (attractors), and some of them may not be accessible by stimulation of a primary sensory area. Note that distinct attractor states are defined here in terms of their spatial patterns, which does not depend on the number of selective excitatory neural pools per area. We developed a numerical approach to identify and count distinct attractors (see Appendix 2 for further details). Our aim is not to exhaustively identify all possible attractors, as the activity space is too large, but to gain insight on how our estimations depend on relevant parameters such as the global coupling strength G, or the maximum area-specific synaptic strength $J_{max}$. Five examples of different distributed WM attractors are shown in *Figure 5A*, where we can appreciate that not all distributed attractors engage cortical areas at all lobes, and that frontal areas are the ones more commonly involved.

A more detailed analysis included four cases depending on the value of the maximum area-specific synaptic strength $J_{max}$ assumed: two of the cases had $J_{max}$ above the bifurcation threshold for isolated areas (0.4655), and the other two had $J_{max}$ below the bifurcation threshold. For the first two cases, having $J_{max} > 0.4655$ means that at least certain areas high in the hierarchy, such as dlPFC, have strong enough local reverberation to sustain activity independently (i.e. they were 'intrinsically multistable' and able to display bistability even when isolated from the network, *Figure 5B*); however, areas lower in the hierarchy like 24 c and F2 would require long-range support to participate in WM. For the last two cases, in which $J_{max} < 0.4655$, none of the areas was able to display bistability when isolated, but they can contribute to stabilize distributed WM attractors as in *Figure 2*. In all four cases, the number of attractors turns out to be an inverted-U function of the global coupling strength G, with an optimal G value maximizing the number of attractors (*Figure 5C*, curves are normalized to have a peak height of one for visualization purposes). This reflects the fact that a minimal global coupling is needed for

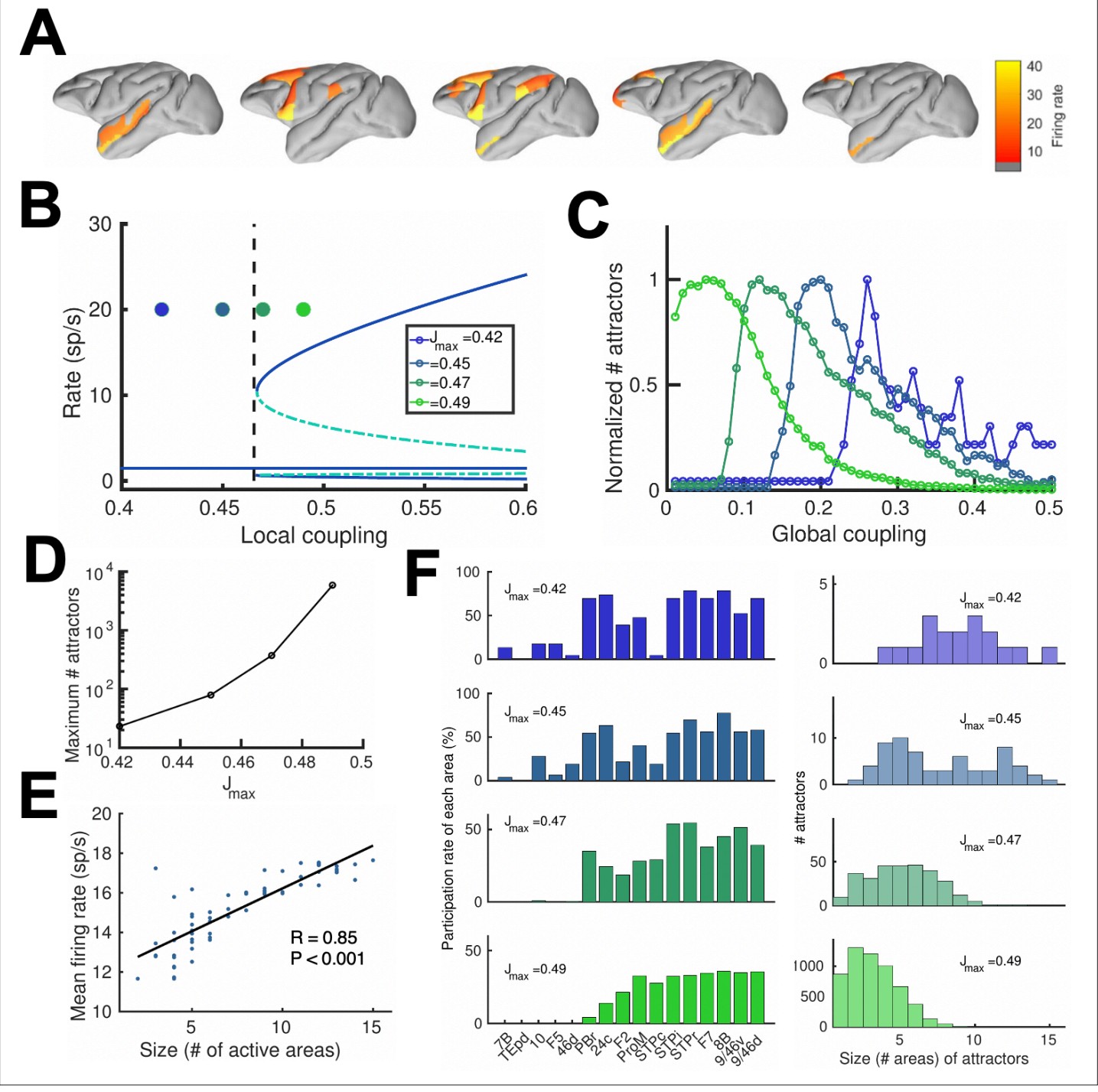

**Figure 5.** Distributed and local WM mechanisms can coexist in the full model. (**A**) Five example distributed attractors of the network model ($J_{max}$ = 0.42). (**B**) Bifurcation diagram of an isolated local area with the four cases considered. (**C**) Number of attractors (normalized) found via numerical exploration as a function of the global coupling for all four cases. (**D**) Maximum (peak) number of attractors for each one of the cases. (**E**) Correlation between size of attractors and mean firing rate of its constituting areas for $J_{max}$ = 0.45 and G = 0.2. (**F**) Participation index of each area (left, arranged by spine count) and distribution of attractors according to their size (right).

areas to coordinate and form distributed WM attractors, but for values of G too large, all areas will follow the majority rule and the diversity and number of possible attractors will decrease. The optimal G value shifted towards lower values for increasing $J_{max}$, and the peak number of attractors simultaneously increasing (***Figure 5D***).

Across all four cases and G values considered, we found a significant positive correlation between the number of areas involved in a given attractor and the average firing rate of these areas (*Figure 5E*), which constitutes an experimentally testable prediction of the distributed model of WM. We also analyzed how distributed WM attractors were constituted for the four different cases (*Figure 5F*). When the network has a high number of intrinsically multistable areas (i.e. when $J_{max}>0.4655$), attractors tend to only involve these areas and are therefore largely restricted to the areas located at the top of the hierarchy (*Figure 5F*, bottom left and right panels). On the other hand, when the network has zero or a low number of intrinsically multistable areas (i.e. $J_{max}<0.4655$), attractors typically involve a larger number of areas (as a larger pool of areas is needed to sustain distributed WM attractors, see top right panel in *Figure 5F*) and the areas involved are more diverse in their composition (*Figure 5F*, top left panel).

## Effects of inactivating areas on distributed attractors

To continue probing the robustness of distributed WM patterns, we tested the effect of inactivating cortical areas in our model during WM tasks, which can be done experimentally using optogenetic methods or lesioning selected areas. We tested this by completely and permanently suppressing the firing rate of the inactivated areas in the model, in such a way that the area becomes a sink of current and does not communicate with other areas. We began by inactivating (or silencing) a given number of randomly selected areas in a visually evoked distributed WM attractor, and found that the number of active areas in the attractor decreases only linearly with the number of inactivated areas (*Figure 6A*). Furthermore, the activity of the areas remained in the distributed WM patterns linearly decreased their sustained activity level with the number of inactivated areas (*Figure 6B*). As silencing areas at the top of the hierarchy could in principle have strong effects, we then systematically silenced areas in reverse hierarchical order (i.e., silencing the top area first, then the top and second-from-top areas, etc), instead of in random order. In this case, the number of active areas decreases a bit more abruptly (*Figure 6C*) and, as we will see later, can prevent the emergence of distributed WM altogether if $J_{max}$ is not sufficiently large.

We also carried out a more systematic evaluation of the effect of cortical inactivations, including their effect on attractors that were not accessible from sensory stimulation directly. This study revealed that inactivating most areas has only limited consequences on the total number of available distributed attractors, although in general the impact increases with the location of the silenced area in the hierarchy (*Figure 6D*). In particular, the overall impact was large when some temporal and prefrontal areas are silenced, and sometimes more than half of the initially available attractors were lost (*Figure 6E*). Interestingly, and beyond any hierarchical dependence, the temporal and prefrontal areas that had the strongest impact are part of a subset of the anatomical network which has a very high (92%) density of connections between its nodes. This anatomical core, which has sparser connections with the remaining areas (forming the periphery of the network) is known as the anatomical 'bowtie hub' of the macaque cortex identified in anatomical studies (*Markov et al., 2013*; *Figure 6F*). Overall, silencing areas at the center of the bowtie had a deeper impact, in terms of the number of attractors surviving the silencing, than silencing areas on the periphery (*Figure 6G*).

## Effects of inactivations and distractors in distributed vs localized WM patterns

Across all analyses performed above, we assumed a relatively large value for the maximum area-specific recurrent strength $J_{max}$ = 0.42, even if still below the critical value needed for bistability in isolation (0.4655). In order to provide clean predictions linked to the distributed WM scenario, in the following sections we studied the case of a strongly distributed WM system with $J_{max}$ = 0.26 and G = 0.48, and compared it to the case of networks which rely purely on a localized WM strategy (with $J_{max}$ = 0.468, G = 0.21 and feedback projections removed to avoid long-range loops).

We first reexamined the effect of inactivations for this strongly distributed WM network. We found that inactivations have in general a stronger effect here than for networks with larger $J_{max}$ (as in *Figure 6*). For example, inactivating key prefrontal areas such as 9/46d (dlPFC) fully prevented the emergence of distributed WM patterns evoked by external stimulation (*Figure 7A and b*), which is in agreement with classical prefrontal lesion studies –see (*Curtis and D'Esposito, 2004*) for a review and a discussion of the implications for dlPFC organization. On the other hand, other areas can still be

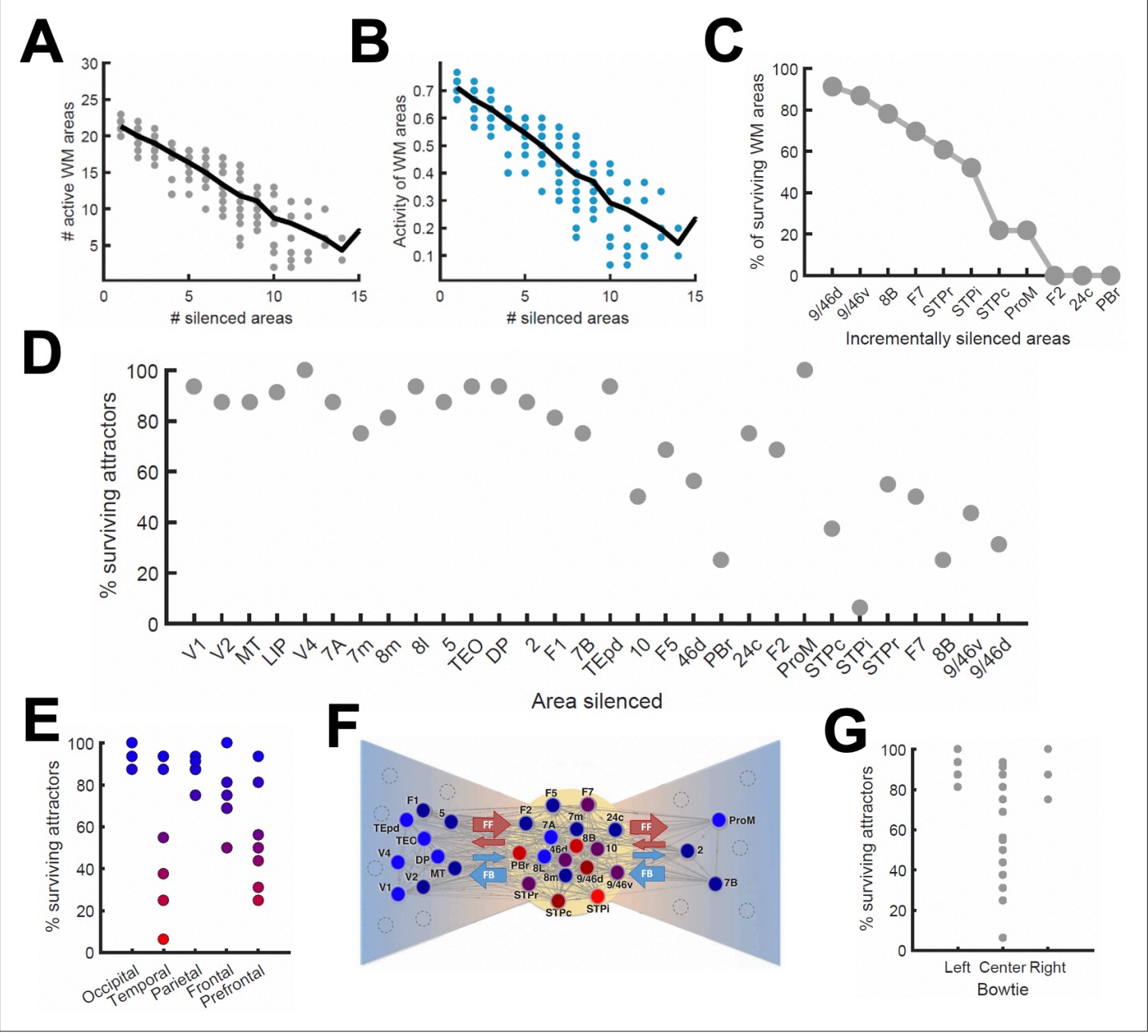

**Figure 6.** Effects of lesioning/silencing areas on the activity and number of attractors. Silencing occurs throughout the full trial for each area indicated here. (**A**) Number of active areas in the example attractor as a function of the number of (randomly selected) silenced areas. (**B**) The activity of the areas which remain as part of the attractor decreases with the number of silenced areas. (**C**) The number of active WM areas decreases faster when areas are incrementally and simultaneously silenced in reverse hierarchical order. (**D**) When considering all accessible attractors for a given network (G = 0.48, $J_{max}$ = 0.42), silencing areas at the top of the hierarchy has a higher impact on the number of surviving attractors than silencing bottom or middle areas. (**E**) Numerical exploration of the percentage of surviving attractors for silencing areas in different lobes. (**F**) Silencing areas at the center of the 'bowtie hub' has a strong impact on WM (adapted from *Markov et al., 2013*). (**G**) Numerical impact of silencing areas in the center and sides of the bowtie on the number of surviving attractors. For panels (**E**) and (**F**), areas color-coded in blue/red have the least/most impact when silenced, respectively.

inactivated without disrupting distributed WM. In some cases, inactivating specific areas might even lead to a disinhibition of other areas and to a general reinforcement of the attractor (e.g. inactivating 24c leads to a larger and faster response by area STPi, *Figure 7B*). This is a consequence of the hierarchical relationship of cortical areas and the counterstream inhibitory bias –silencing a top area which

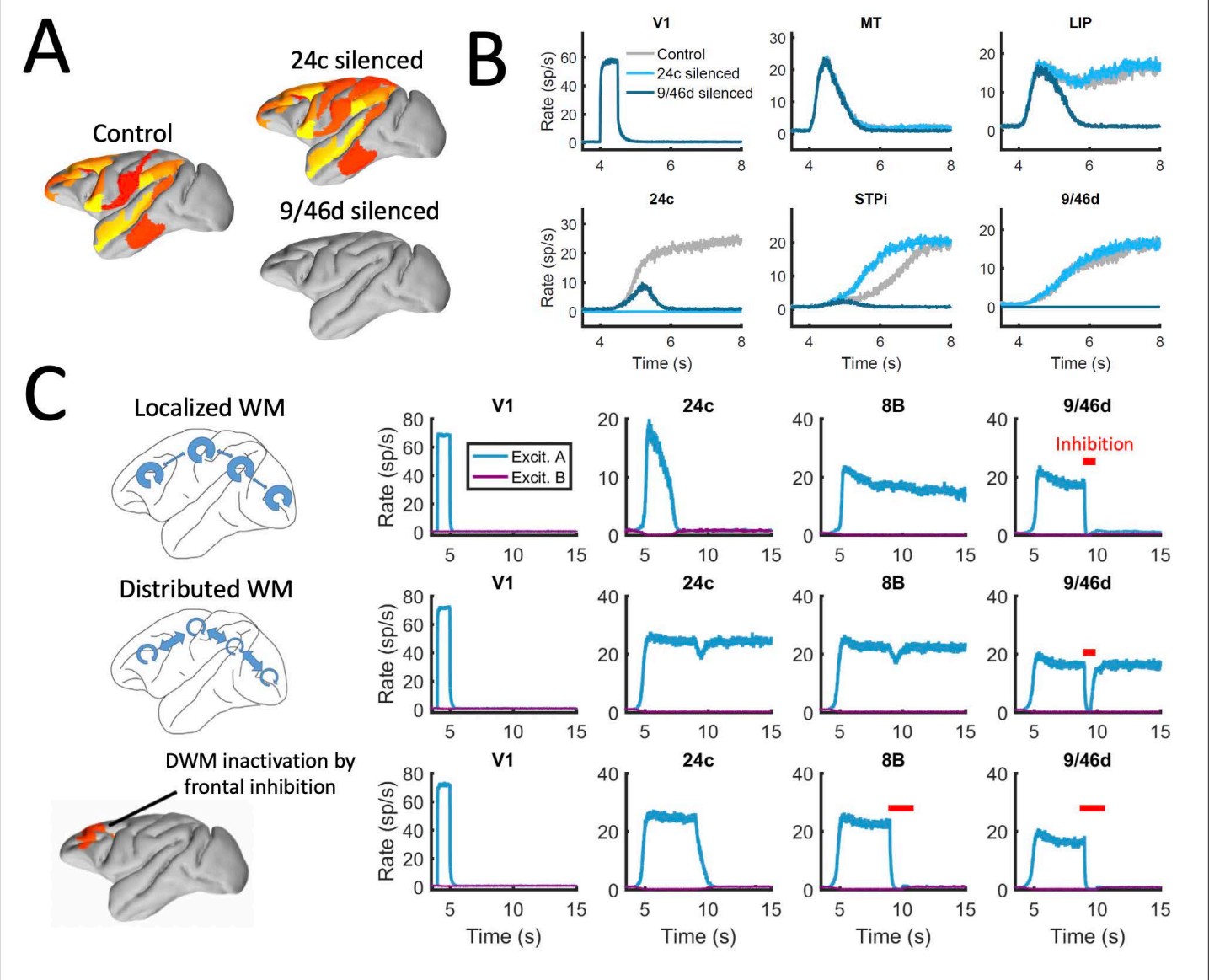

**Figure 7.** Effect of silencing areas in localized vs distributed WM. (**A**) Full-brain activity maps during the delay period for the control case (left), and lesioning/silencing area 24 (top right) or area 9/46d (bottom right). (**B**) Traces of selected areas for the three cases in panel A show the effects of silencing each area. (**C**) For a network displaying localized WM (top row, corresponding to $J_{max} = 0.468$, $G = 0.21$), a brief inactivation of area 9/46d leads to losing the selective information retained in that area. For a network displaying distributed working memory (middle row, $J_{max} = 0.26$, $G = 0.48$) a brief inactivation removes the selective information only transiently, and once the external inhibition is removed the selective information is recovered. In spite of this robustness to brief inactivations, distributed WM patters can be shut down by inhibiting a selective group of frontal areas simultaneously (bottom row, inhibition to areas 9/46 v, 9/46d, F7, and 8B). The shut-down input, of strength I = 0.3 and 1s duration, is provided to the nonselective inhibitory population of each of these four areas.

is effectively inhibiting lower areas might release these lower areas from the inhibition and increase their firing.

In addition to permanently inactivating areas, we tested the effects of brief (500ms ~ 1 s) inactivations in specific areas, and compare the effects in localized vs distributed WM scenarios. For networks relying on localized WM, areas at the top of the hierarchy maintained their selective information largely independent from each other. Consequently, briefly inactivating area 9/46d would not, for example, have an effect on the sustained activity of other areas such as 8B (*Figure 7C*, top row). Furthermore, the brief inactivation was enough to remove the information permanently from 9/46d, which remained in the spontaneous state after the inhibition was withdrawn. On the other hand, silencing an area like

9/46d will slightly affect the sustained activity in other areas (such as 8B) in a network strongly relying on distributed WM (*Figure 7C*, middle row). However, area 9/46d will be able to recover the encoded selective information once the inhibitory pulse is removed, due to the strong interaction between cortical areas during the delay period. This constitutes a strong prediction for networks which rely on distributed interactions to maintain WM.

The marked resilience of distributed WM attractors to brief inhibitory pulses raises the question of how to shut down the sustained activity once the task has been done. In many traditional WM models, this is achieved by providing a strong nonspecific excitatory input to the whole WM circuit, which triggers inhibition and drives the activity of the selective populations back to their spontaneous state (*Compte, 2006*; *Compte et al., 2000*; *Wang, 1999*). It is, however, unrealistic to expect that this approach could also be used for shutting down distributed WM patterns, as it would require a large-scale synchronous inhibitory pulse to all active areas.

We therefore explore in our model whether more spatially selective signals can shut down distributed patterns of activity. In spite of their robustness to sensory distractors as discussed above, we find that distributed WM activity patterns can be shut down with an excitatory input targeting inhibitory populations of areas high in the hierarchy. *Figure 7C* (bottom row) shows how a visually evoked distributed WM attractor is deactivated when we deliver excitatory input to the inhibitory populations in the top four areas of the hierarchy (9/46v, 9/46d, F7 and 8B). These prefrontal areas are spatially localized and thought to be highly engaged in WM maintenance, and therefore they are suitable candidates to control the suppression of sustained activity in other cortical areas, such as areas LIP and

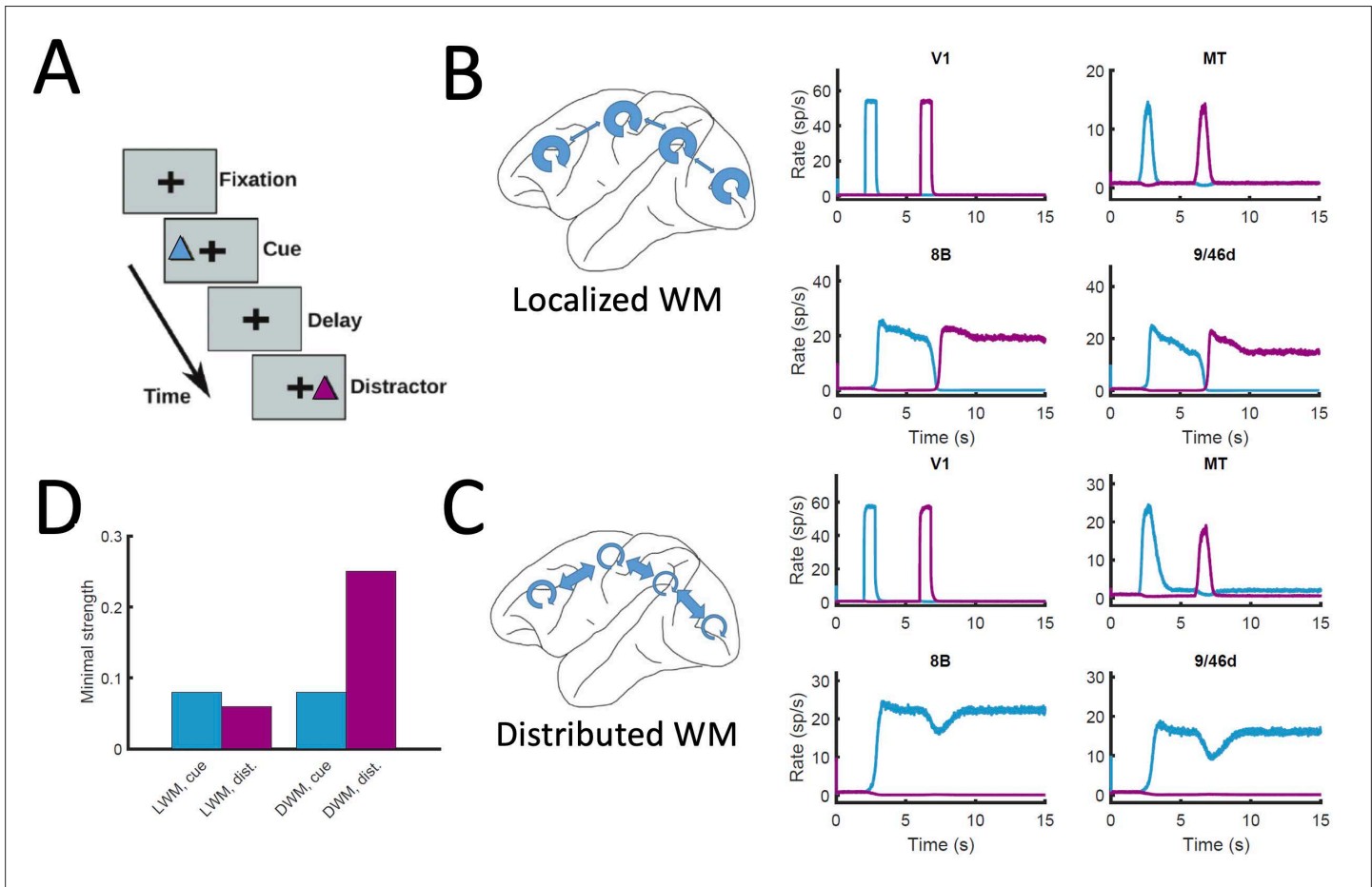

**Figure 8.** Resistance to distractors in localized vs distributed WM. (**A**) Scheme of the WM task with a distractor, with the cue (current pulse of strength $I_A$ = 0.3 and duration 500ms) preceding the distractor ($I_B$ = 0.3, 500ms) by four seconds. (**B**) Activity traces of selected areas during the task, for a network displaying localized WM ($J_{max}$ = 0.468, G = 0.21). (**C**) Same as panel B, but for a model displaying distributed WM ($J_{max}$ = 0.26, G = 0.48). (**D**) Minimal strength required by the cue (blue) to elicit a sustained activity state, and minimal strength required by the distractor (purple) to remove the sustained activity, both for localized WM (left) and distributed WM (right).

24c. Therefore, in spite of engaging cortical areas across all four lobes, distributed WM attractors can be controlled and deactivated by localized inhibition to a small set of frontal areas.

Finally, the distributed nature of WM has also implications for the impact of distractors and similar sensory perturbations on maintenance of selective activity and overall performance of the network. We simulated a delayed response task with distractors (*Figure 8A*), in which stimulus A is cued to be maintained in WM and stimulus B is presented as distractor during the delay period (and vice versa). When simulated in the localized WM network, we observed that distractors with the same saliency than the original cue were sufficient to switch the network into the new stimuli, making the network easy to distract (*Compte et al., 2000*; *Figure 8B*). For the case of distributed WM, however, the network was highly resilient, and distractors with similar saliency levels as the input cues were filtered out by the network so that working memory storage is preserved (*Figure 8C*). Overall, we found that localized WM networks can be distracted with stimuli similar or even weaker than the minimum cue input strength required to encode a WM pattern, while effective distractors need to be about three times as strong in the case of distributed WM networks (*Figure 8D*). This difference is due to the robustness of a distributed attractor compared to a local circuit mechanism, but also to the effect of the counterstream inhibitory bias which dampens the propagation of distractor signals (cf. MT responses in *Figure 8B and C*). This constitutes a key difference between distributed and local WM models feasible of experimental validation.

## Discussion

The investigation of cognitive functions has been traditionally restricted to operations in local brain circuits –mostly due to the limitations on available precision recording techniques to local brain regions, a problem that recent developments are starting to overcome (*Jun et al., 2017*; *Panichello and Buschman, 2021*; *Siegel et al., 2015*; *Stringer et al., 2019*). It is therefore imperative to advance in the study of distributed cognition using computational models as well, to support experimental advances. In this work, we have presented a large-scale circuit mechanism of distributed working memory, realized by virtue of a new concept of robust transition in space. The distributed WM scenario is compatible with recent observations of multiple cortical areas participating in WM tasks (*Christophel et al., 2017*; *Leavitt et al., 2017*; *Sreenivasan and D'Esposito, 2019*), even when some of these areas have not been traditionally associated with WM. Importantly, considering distributed WM in a truly large-scale network has revealed phenomena such as the transition in cortical space (*Figure 2D*), the counterstream inhibition (*Figure 4*) and the diverse pool of available attractors (*Figure 5*) which would not emerge when studying systems of one or two cortical areas as in *Edin et al., 2009*; *Guo et al., 2017*; *Murray et al., 2017b*.

One of the main ingredients of the model is the gradient of excitation across the cortical hierarchy, implemented via an increase of excitatory recurrent connections (hinted by the existing anatomical evidence on dendritic spines on pyramidal cells across multiple cortical areas *Elston, 2007*). The particular values of projection strengths do not impact the emergence of distributed WM patterns, but they influence the performance of individual areas (*Figure 4—figure supplement 2E-H*). Moreover, we introduce the concept of counterstream inhibitory bias (CIB) which was found to stabilize distributed yet spatially confined mnemonic sustained activity patterns in spite of dense inter-areal connectivity. Evidence compatible with CIB includes anatomical studies of feedback projections targeting supragranular layers (*Markov et al., 2014a*; *Rockland and Van Hoesen, 1994*), which contain multiple types of inhibitory neurons, and electrophysiological studies showing that figure-ground segregation requires at least partially inhibitory feedback (*Hupé et al., 1998*). CIB is also compatible with other theoretical frameworks such as predictive coding, which require inhibitory feedback to minimize prediction errors along the cortical hierarchy (*Bastos et al., 2012*).

Macroscopic gradients and hierarchical structures have recently been proposed as a general principle for understanding heterogeneities in the cortex (*Wang, 2020*). A standing challenge is to clarify how structural gradients relate to functional ones –for example, the gradual progression from sensory-related to task-related activity as one ascends in the cortical hierarchy, or the higher mixed selectivity in higher cortical areas. For example, it has been shown that gradients of circuit properties in line with hierarchical structures contribute to the emergence of a gradient of time scales across cortex, supporting slow dynamics in prefrontal areas (*Chaudhuri et al., 2015*; *Murray et al., 2014*; *Wang, 2020*) (see also *Figure 4—figure supplement 3*), and also that a hierarchical organization of

functional states could serve as basis for WM-guided decisions and executive control (*Miller et al., 2018*; *Muhle-Karbe et al., 2020*). It is possible that structural gradients would play a role not only in other cognitive functions in monkeys, but also in other animals including mice (*Fulcher et al., 2019*) and humans (*Burt et al., 2018*; *Demirtaş et al., 2019*).

Theoretically, the present work is the first to show that graded changes of circuit properties along the cortical hierarchy provides a mechanism to explain qualitatively distinct functions of different cortical areas (whether engaged in working memory). This is reminiscent of the phenomenon mathematically called bifurcation, which denotes the emergence of novel behavior as a result of quantitative property change as a control parameter in a nonlinear dynamical system (*Strogatz, 1994*). Our model displays a novel form of transition across the cortex, which cannot be simply explained by a parameter change laid out spatially by virtue of a macroscopic gradient, because areas are densely connected with each other in a complex large-scale network. Such a transition implies that a few association areas should exhibit signs of dynamical criticality akin to water near a transition between gas and liquid states. This will be explored further in the future.

Interestingly, the model uncovered a host of distinct sustained activity attractor states, each with its own transition location in the cortical tissue. They are defined by their spatial distributed patterns in the large-scale cortical system, independent of the number of selective neural pools per area (*Figure 5A*). Many of these mnemonic activity states are not produced by stimulation of primary sensory areas. These attractor internal states could serve various forms of internal representations such as those that are not triggered by a particular sensory pathway –or those triggered by sensory input but are encoded differently as memories (*Mendoza-Halliday and Martinez-Trujillo, 2017*). The identification of these internal representations in further detail are beyond the scope of the present study, but uncovering their functional role should be within reach of additional experimental and computational work.

Although our proposal extends the idea of attractor dynamics to the scale of large networks, there are several fundamental differences between our model and standard Hopfield-like models of local attractor dynamics. In a large network in which the areas share a preferred selectivity (i.e. the population 'A' in *Figure 1B*), a Hopfield model would trigger a sustained activity of all the populations selective to 'A' across the network, which is incompatible with experimental observations (*Leavitt et al., 2017*). More complex patterns with early sensory areas inactive can be learned by Hopfield models, but only at the expense of modifying the selectivity of population 'A' in those areas. On the other hand, our model considers the gradient of properties as a partial solution to this problem, and even tests the validity of such solutions for realistic local coupling levels, which in turn leads to the prediction of the CIB. Overall, our model constitutes an example of how classical ideas of local circuit dynamics may be translated to large-scale networks, and the corresponding new theoretical insights that such process brings along.

## Extending the model of distributed working memory

The reported model of large-scale cortical networks is, to the best of our knowledge, the first of its kind addressing a cardinal cognitive function in a data-constrained way, and it opens the door for elucidating this and similar complex brain processes in future research. Several avenues may be taken to extend the functionality of the present model. First, it is straightforward to have an arbitrary number of selective neural pools per area (*Wang, 1999*), which would increase both the selectivity to sensory inputs and the available number of distributed WM attractors. In that case, more complex connections (not necessarily A to A, B to B, etc.) can be investigated, including a distance-dependent 'ring structure' (*Compte et al., 2000*) or random connections (*Bouchacourt and Buschman, 2019*). Second, the model presented here is limited to 30 cortical areas, and can be expanded to include both additional cortical areas and subcortical structures relevant for working memory such as thalamic nuclei *Guo et al., 2017*; *Jaramillo et al., 2019* as their connectivity data become available. Interesting extensions in this sense could involve mouse connectomics, to explore the role of thalamocortical loops (*Guo et al., 2017*) in sustained activity (although working memory mechanisms could differ between rodents and primates), and human connectomics, to reveal the potential influence of complex network structures in the emergence of distributed distractors (*Bassett and Sporns, 2017*; *Demirtaş et al., 2019*; *van den Heuvel and Sporns, 2013*). Third, electrophysiological recording from multiple brain areas could be used to further constrain the dynamics of the model. For example,

when extending the model to more complex WM tasks involving components of attention or sensorimotor decisions, additional electrophysiological data could improve the model's predictive power, especially for areas such as V4 or LIP (*Panichello and Buschman, 2021*; *Siegel et al., 2015*). Fourth, the model can be improved by incorporating more biological details such as cortical layers (*Mejias et al., 2016*), contributions of different neuromodulators, and various types of inhibitory neurons.

## Attractor model of working memory and activity-silent state models

In the description adopted here, we have considered that working memory maintained via selective activity is described as an attractor state (*Amit and Brunel, 1997*; *Compte, 2006*; *Wang, 2001*). Other mechanisms have also been proposed, including the maintenance of memories via feedforward mechanisms and activity-silent state mechanisms (*Barbosa et al., 2020*; *Goldman, 2009*; *Kamiński and Rutishauser, 2020*; *Miller et al., 2018*; *Mongillo et al., 2008*; *Stokes, 2015*; *Trübutschek et al., 2017*; *Wolff et al., 2017*). Importantly, not limited to steady-states, the attractor framework is fully consistent with temporal variations of delay activity. For instance, during a mnemonic delay a working memory circuit can exhibit stochastic oscillations in the gamma (~40Hz) frequency range, in which neurons often stop momentarily before resuming spike firing, the temporal gap of silence is bridged by slow synaptic reverberation (*Compte et al., 2000*; *Lundqvist et al., 2016*). Another example is self-sustained repetition of brief bursts of spikes interspersed with long silent time epochs (*Mi et al., 2017*). As discussed in a recent review (*Wang, 2021*), the real conceptual alternative to attractor states is transient activity that fades away in time while a memory trace remains as a hidden state. The biological mechanisms such as the NMDA receptors at recurrent excitatory synapses or short-term synaptic plasticity are not fundamentally separate. A stable (attractor) state does not mean the absence of short-term synaptic facilitation which, as an activity-dependent process can contribute to the maintenance of an attractor state by supplying sufficient excitatory reverberation (*Hempel et al., 2000*; *Pereira and Wang, 2015*; *Mi et al., 2017*), enhance robustness of self-sustained mnemonic activity (*Hansel and Mato, 2013*; *Itskov et al., 2011*; *Mejias and Torres, 2009*; *Mejias et al., 2012*; *Pereira and Wang, 2015*; *Seeholzer et al., 2019*) and induce cross-trial serial effects (*Barbosa et al., 2020*; *Bliss et al., 2017*). When the combined strength of excitatory-to-excitatory connections and short-term plasticity is sufficient to maintain a mnemonic state, it is mathematically described as an attractor, no matter how complex its spatiotemporal dynamics may be; otherwise, there is not enough reverberation and neural firing would decay over time over a sufficiently long delay period and never returns spontaneously (*Mongillo et al., 2008*). Strictly speaking, only the latter should be referred to as activity-silent. An activity-silent state also depends on spiking activity to refresh hidden memory traces and to readout the stored information content. Short-term plasticity could therefore contribute to activity-silent memory traces but also to self-sustained activity. It is worth noting that the above discussion is limited to a local circuit model. The assumed absence of any input to it is unlikely to hold in real life, when there are always external stimulation from the environment and internal processes (e.g. intruding thoughts) in other brain regions that project to the local area under consideration. A mixture of activity-silent state and episodic spikes caused by inputs from the rest of the brain represents an interesting possibility that a local network model is not suitable to investigate. Multi-regional modeling as reported here in interplay with new experiments in the future will shed insights into such a scenario.

Multi-regional network modeling should be extended to explore complex dynamics and cell-to-cell heterogeneity of neural population activity patterns underlying working memory representations (*Druckmann and Chklovskii, 2012*; *Hussar and Pasternak, 2009*; *Lim and Goldman, 2013*; *Murray et al., 2017a*; *Stokes, 2015*; *Wimmer et al., 2016*). There is no reason to think that the encoding of memory items could not use the complex spatiotemporal interactions between brain areas instead of just local interactions. A large-scale implementation of WM also pairs well with recent hypotheses in which memory selectivity is reached via dynamical flexibility instead of content-based attractors, since the wide number and heterogeneity of long-range projections would reduce connection overlap and alleviate the limit capacity of these models (*Bouchacourt and Buschman, 2019*). The use of inter-areal interactions to sustain WM-related activity has been explored in other recent works (*Edin et al., 2009*; *Guo et al., 2017*; *Murray et al., 2017b*); however, this was limited to two-area systems and the models were not anatomically constrained, therefore limiting their predictive power. Frameworks of WM in which oscillations play an active role, for example regarding WM-guided executive control

(*Miller et al., 2018*), may benefit from using distributed WM approaches, given the usefulness of previous models of large-scale brain networks to explain oscillatory phenomena in the macaque brain (*Mejias et al., 2016*). Finally, with a simplified model we show that, when short-term facilitation is incorporated at an appropriate level, it enhances the synaptic efficacy in areas at the top of the hierarchy. A more extended consideration of short-term synaptic plasticity, and the contrast between self-sustained activity versus activity-silent state was reported elsewhere (*Froudist-Walsh et al., 2021*).

## Experimental predictions provided by our model

The distributed WM model presented here yields four experimentally testable predictions in monkey (and potentially rodent) experiments, which can be used to validate our theory. First, the model predicts a positive correlation between the number of areas involved in a WM task and their average firing rate of sustained activity (*Figure 5E*). Such relationship should not occur according to models of localized WM, since activity levels would be fairly independent across areas. Only a distributed WM model in which neurons of similar selectivity (but located at different areas) support each other via long-range projections would lead to this prediction. This prediction could be tested with neuroimaging experiments, by correlating the level of activation of different brain regions during WM with the number of regions activated. Existing data could be used to carefully test this prediction in future studies. A complementary version of this prediction is that, if areas displaying sustained activity are silenced (e.g. optogenetically), the activity of the other sustained activity areas will decrease (*Figure 6B*).

Second, our model predicts that areas involved in distributed WM patterns can be briefly silenced without losing the encoded information, which will be recovered as soon as the inhibition is gone (*Figure 7*, middle row), something that localized WM do not predict (*Figure 7* top row; see however, related effects on continuous attractor models *Seeholzer et al., 2019*). As in the first prediction, large-scale interactions across neurons of similar selectivity are a condition for this phenomenon, according to our model. Optogenetic inactivations could be used to test this result.

Third, distributed WM is significantly more robust to distractors than localized WM (*Figure 8*), due to their intrinsic resilience and the inhibitory feedback condition. Behavioral and neuroimaging experiments in macaques should be able to test this, by testing potential correlations between the spatial extension of a distributed WM pattern and its robustness of the corresponding trials to distractors.

Fourth, electrophysiological recordings in macaques could test whether FEF areas require support from frontal areas (in the form of strong excitation) to maintain WM-related activity (*Figure 4—figure supplement 1*). In particular, coactivation between FEF and frontal areas could be correlated with elevated activity in FEF neurons. Although this prediction focuses on a particular set of areas, it should shed light into unclear aspects of FEF dynamics.

In a more general sense, our model predicts a reversed gradient of inhibition and strong large-scale interactions to sustain distributed WM patterns, which may be observed using different experimental approaches. It will also be interesting to see whether the same model is able to account for decision-making processes as well as working memory (*Wang, 2002*; *Wong and Wang, 2006*).

Given that our model is constrained using data from the macaque brain, it is interesting to discuss which of our results would extend to other conditions (and, in particular, to other animal models). First, we have shown with the simplified model (*Figure 3*) that the emergence of distributed WM requires minimal elements and is therefore likely to emerge in cortical networks of other animals such as rodents or humans. The existence of a CIB (*Figure 4*) is a requirement for distributed WM as long as the gradient of local properties grows sublinearly, which makes the CIB a plausible condition in other species as well. Our results on the activity and number of attractors (*Figure 5*) and the effects of silencing (*Figures 6 and 7*) are highly dependent on the anatomical constraints used, so their validity will need to be tested for rodents and humans. Finally, the results on the robustness to distractors (*Figure 8*) rely on the presence of distributed activity (with moderated values of local coupling strengths) and the effect of the CIB on incoming distractor signals, so as long as these ingredients are present in other species, we should expect these effects to be there as well.

Conceptually, this work revealed a novel mechanism in cortical space to generate differential functions across different cortical areas, a concept that is likely to be generalizable for understanding how distinct cortical areas endowed with a canonical circuit organization are at the same time suited for differential functions (*Wang, 2020*).

## Materials and methods

### Anatomical data

The anatomical connectivity data used has been gathered in an ongoing track tracing study in macaque and has been described in detail elsewhere (*Markov et al., 2013*; *Markov et al., 2014a*; *Markov et al., 2014b*; *Mejias et al., 2016*). Briefly, retrograde tracer injected into a given target area labels neurons in a number of source areas projecting to the target area. By counting the number of labeled neurons on a given source area, Markov et al. defined the fraction of labeled neurons (FLN) from that source to the target area. FLN can serve as a proxy for the 'connection strength' between two cortical areas, which yields the connectivity pattern of the cortical network (*Figure 1—figure supplement 1A-B*). In addition, Markov et al. also measured the number of labeled neurons on the supragranular layer of a given source area. Dividing this number over the total number of labeled neurons on that area, we can define the supragranular layered neurons (SLN) from that source to the target area (*Figure 1—figure supplement 1C-D*).

SLN values may be used to build a well-defined anatomical hierarchy (*Felleman and Van Essen, 1991*; *Markov et al., 2014b*). Source areas located lower (higher) than the target area in the anatomical hierarchy, as defined in *Felleman and Van Essen, 1991*, display a progressively higher (lower) proportion of labeled neurons in the supragranular layer. As a consequence, the lower (higher) the source area relative to the target area, the higher (lower) the SLN values of the source-to-target projection. By performing a logistic regression on the SLN data to accommodate each area in its optimal position in the anatomical hierarchy (*Chaudhuri et al., 2015*), we assign a hierarchical value $h_i$ to each area 'i'.

Iterating these measurements across other anatomical areas yields an anatomical connectivity matrix with weighted directed connections and an embedded structural hierarchy. The 30 cortical used to build our data-constrained large-scale brain network are, in hierarchical order: V1, V2, V4, DP, MT, 8 m, 5, 8 l, 2, TEO, F1, STPc, 7 A, 46d, 10, 9/46 v, 9/46d, F5, TEpd, PBr, 7 m, LIP, F2, 7B, ProM, STPi, F7, 8B, STPr and 24 c. Finally, data on wiring connectivity distances between cortical areas is available for this dataset as well, allowing to consider communication time lags when necessary (we found however that introducing time lags this way does not have a noticeable impact on the dynamics of our model). The connectivity data used here is available to other researchers from https://core-nets.org.

The corresponding 30 × 30 matrices of FLN and SLN are shown in *Figure 1—figure supplement 1B, D*. Areas in these matrices are arranged following the anatomical hierarchy, which is computed with the SLN values and a generalized linear model (*Chaudhuri et al., 2015*; *Mejias et al., 2016*). Surgical and histology procedures followed European requirements 86/609/EEC and were approved by the ethics committee of the Rhone-Alpes region.

In addition to the data on FLN and SLN across 30 cortical areas, we used additional data to constrain the area-to-area differences in the large-scale brain network. In particular, we have collected data on the total spine count of layer 2/3 pyramidal neuron basal dendrites across different cortical areas, as the spine count constitutes a proxy for the density of synaptic connections within a given cortical area (*Elston, 2007*). A full list of all area-specific values of spine densities considered and their sources is given in *Table 1*. We use an age correction factor meant to correct for the decrease of spine counts with age for data obtained from old monkeys. A plausible estimate would be a ~ 30% decrease for a 10y difference (*Duan et al., 2003*; *Young et al., 2014*). See *Figure 1—figure supplement 2* for the effect of this correction on the overall gradient established by the spine count data, and the correlation of such gradient with the SLN hierarchy.

### Experimental evidence of WM-related activity across cortical areas

To compare the results of our model with existing evidence, we generated brain maps highlighting areas for which experimental evidence of WM-related activity during the delay period has been found. Following the data collected by recent review studies (*Christophel et al., 2017*; *Leavitt et al., 2017*), we distinguish between three categories. First, areas with strong WM evidence (for which at least two studies show support of WM-related activity, or if only studies supporting WM activity are known) are shown in dark blue in the maps of *Figures 2 and 4*. Second, areas with moderate evidence (for which substantial positive and negative evidence exist) are shown in light blue. Finally, areas for which strong negative evidence exists (more than two studies with negative evidence, or absence of any positive

**Table 1.** Spine count data from basal dendrites of layer 2/3 pyramidal neurons in young (~2y o) macaque, acquired from the specified literature.

See also *Figure 1—figure supplement 2*.

| Rank in SLN hierarchy | Area name | Measured spine count | Age correction factor | Source |
|---|---|---|---|---|
| 1 | V1 | 643 | 1 | *Elston et al., 1999*; *Elston and Rosa, 1997* |
| 2 | V2 | 1201 | 1 | *Elston and Rosa, 1997* |
| 3 | V4 | 2429 | 1 | *Elston and Rosa, 1998b* |
| 4 | DP | - | - | |
| 5 | MT | 2077 | 1 | *Elston et al., 1999* |
| 6 | 8m | 3200 | 1.30 | *Elston and Rosa, 1998b* |
| 7 | 5 | 4689 | 1 | *Elston and Rockland, 2002* |
| 8 | 8l | 3200 | 1.30 | *Elston and Rosa, 1998b* |
| 9 | 2 | - | - | |
| 10 | TEO | 4812 | 1 | *Elston and Rosa, 1998b* |
| 11 | F1 | - | - | |
| 12 | STPc | 8337 | 1 | *Elston et al., 1999* |
| 13 | 7a | 2572 | 1 | *Elston and Rosa, 1997*; *Elston and Rosa, 1998a* |
| 14 | 46d | 6600 | 1.15 | Estimated from *Elston, 2007*; |
| 15 | 10 | 6488 | 1.15 | *Elston et al., 2011* |
| 16 | 9/46 v | 7800 | 1.15 | Estimated from *Elston, 2007* |
| 17 | 9/46d | 7800 | 1.15 | Estimated from *Elston, 2007* |
| 18 | F5 | - | - | |
| 19 | TEpd | 7260 | 1 | *Elston et al., 1999* |
| 20 | PBr | - | - | |
| 21 | 7m | 2294 | 1.30 | *Elston, 2001* |
| 22 | LIP | 2316 | 1 | *Elston and Rosa, 1997*; *Elston and Rosa, 1998a* |
| 23 | F2 | - | - | |
| 24 | 7B | 6841 | 1 | *Elston and Rockland, 2002* |
| 25 | ProM | - | - | |
| 26 | STPi | 8337 | 1 | *Elston et al., 1999* |
| 27 | F7 | - | - | |
| 28 | 8B | - | - | |
| 29 | STPr | 8337 | 1 | *Elston et al., 1999* |
| 30 | 24 c | 6825 | 1.15 | *Elston et al., 2005* |

studies) are left as grey in the map. Alternative criteria have only small effects on the resulting maps and the general results are consistent to variations.

## Computational model: local neural circuit

We describe the neural dynamics of the local microcircuit representing a cortical area with the Wong-Wang model (*Wong and Wang, 2006*). In its three-variable version, this model describes the temporal evolution of the firing rate of two input-selective excitatory populations as well as the evolution of the firing rate of an inhibitory population. All populations are connected to each other (see *Figure 1A*). The model is described by the following equations:

$$\frac{dS_A}{dt} = -\frac{S_A}{\tau_N} + \gamma \left(1 - S_A\right) r_A \tag{1}$$

$$\frac{dS_B}{dt} = -\frac{S_B}{\tau_N} + \gamma \left(1 - S_B\right) r_B \tag{2}$$

$$\frac{dS_C}{dt} = -\frac{S_C}{\tau_G} + \gamma_I r_C \tag{3}$$

Here, $S_A$ and $S_B$ are the NMDA conductances of selective excitatory populations A and B respectively, and $S_C$ is the GABAergic conductance of the inhibitory population. Values for the constants are $\tau_N$=60 ms, $\tau_G$=5 ms, $\gamma$ = 1.282 and $\gamma_I$=2. The variables $r_A$, $r_B$ and $r_C$ are the mean firing rates of the two excitatory and one inhibitory populations, respectively. They are obtained by solving, at each time step, the transcendental equation $r_i = \phi_i \left(I_i\right)$ (where $\phi$ is the transfer function of the population, detailed below), with $I_i$ being the input to population 'i', given by

$$I_A = J_s S_A + J_c S_B + J_{EI} S_C + I_{0A} + I_{net}^A + x_A \left(t\right) \tag{4}$$

$$I_B = J_c S_A + J_s S_B + J_{EI} S_C + I_{0B} + I_{net}^B + x_B \left(t\right) \tag{5}$$

$$I_C = J_{IE} S_A + J_{IE} S_B + J_{II} S_C + I_{0C} + I_{net}^C + x_C \left(t\right) \tag{6}$$

In these expressions, $J_s$, $J_c$ are the self- and cross-coupling between excitatory populations, respectively, $J_{EI}$ is the coupling from the inhibitory populations to any of the excitatory ones, $J_{IE}$ is the coupling from any of the excitatory populations to the inhibitory one, and $J_{II}$ is the self-coupling strength of the inhibitory population. The parameters $I_{0i}$ with i = A, B, C are background inputs to each population. Parameters are $J_s$ = 0.3213 nA, $J_c$ = 0.0107 nA, $J_{IE}$ = 0.15 nA, $J_{EI}$ = −0.31 nA, $J_{II}$ = −0.12 nA, $I_{0A}$=$I_{0B}$ = 0.3294 nA and $I_{0C}$=0.26 nA. Later we will modify some of these parameters in an area-specific manner (in particular $J_s$ and $J_{IE}$) to introduce a gradient of properties across the cortical hierarchy. The term $I_{net}^i$ denotes the long-range input coming from other areas in the network, which we will keep as zero for now but will be detailed later. Sensory stimulation can be introduced here as extra pulse currents of strength $I_{pulse}$ = 0.3 and duration $T_{pulse}$ = 0.5 sec (unless specified otherwise).

The last term $x_i(t)$ with i = A, B, C is an Ornstein-Uhlenbeck process, which introduces some level of stochasticity in the system. It is given by

$$\tau_{noise} \frac{dx_i}{dt} = -x_i + \sqrt{\tau_{noise}} \, \sigma_i \, \xi_i \left(t\right) \tag{7}$$

Here, $\xi_i(t)$ is a Gaussian white noise, the time constant is $\tau_{noise}$=2 ms and the noise strength is $\sigma_{A,B}$=0.005 nA for excitatory populations and $\sigma_C$=0 for the inhibitory one.

The transfer function $\phi_i(t)$ which transform the input into firing rates takes the following form for the excitatory populations (*Abbott and Chance, 2005*):

$$\phi_{A,B} \left(I\right) = \frac{aI - b}{1 - \exp\left[-d \left(aI - b\right)\right]} \tag{8}$$

The values for the parameters are $a$ = 135 Hz/nA, $b$ = 54 Hz, and $d$ = 0.308 s. For the inhibitory population a similar function can be used, but for convenience we choose a threshold-linear function:

$$\phi_C \left(I\right) = \left[\frac{1}{g_I} \left(c_1 I - c_0\right) + r_0\right]_+ \tag{9}$$

The notation $\left[x\right]_+$ denotes rectification. The values for the parameters are $g_I$ = 4, $c_1$ = 615 Hz/nA, $c_0$ = 177 Hz and $r_0$ = 5.5 Hz. Finally, it is sometimes useful for simulations (although not a requirement) to replace the transcendental equation $r_i = \phi_i \left(I_i\right)$ by its analogous differential equation, of the form

$$\tau_r \frac{dr_i}{dt} = -r_i + \phi_i \left(I_i\right) \tag{10}$$

The time constant can take a typical value of $\tau_r$=2 ms.

## Computational model: gradient of synaptic strengths

Before considering the large-scale network and the inter-areal connections, we look into the area-to-area heterogeneity to be included in the model.

Our large-scale cortical system consists of N = 30 local cortical areas, for which inter-areal connectivity data is available. Each cortical area is described as a Wong-Wang model of three populations like

the ones described in the previous section. Instead of assuming areas to be identical to each other, here we will consider some of the natural area-to-area heterogeneity that has been found in anatomical studies. For example, work from *Elston, 2007* has identified a gradient of dendritic spine density, from low spine numbers (~600) found in early sensory areas to large spine counts (~9000) found in higher cognitive areas. On the other hand, EPSP have similar values both in early sensory (~1.7 ± 1.3 mV) and higher cognitive areas (~0.55 ± 0.43 mV). The combination of these findings suggests an increase of local recurrent strength as we move from sensory to association areas. In addition, cortical areas are distributed along an anatomical hierarchy (*Felleman and Van Essen, 1991*; *Markov et al., 2014a*). The position of a given area 'i' within this hierarchy, namely $h_i$, can be computed with a generalized linear model using data on the SLN (fraction of supragranular layer neurons) projecting to and from that area. In particular, we assigned hierarchical values to each area such that the difference in values predicts the SLN of a projection. Concretely, we assign a value $H_i$ to each area $A_i$ so that $SLN(A_j \rightarrow A_i) \sim f(H_i - H_j)$, with 'f' being a logistic regression. The final hierarchical values are then obtained by normalizing $h_i = H_i / H_{max}$. Further details on the regression are provided elsewhere (*Chaudhuri et al., 2015*; *Markov et al., 2014b*).

In the following, we will assign the incoming synaptic strength (both local and long-range) of a given area as a linear function of the dendritic spine count values observed in anatomical studies, with age-related corrections when necessary. Alternatively, when spine count data is not available for a given area, we will use its position in the anatomical hierarchy, which displays a high correlation with the spine count data, as a proxy for the latter. After this process, the large-scale network will display a gradient of local and long-range recurrent strength, with sensory/association areas showing weak/ strong local connectivity, respectively. We denote the local and long-range strength value of a given area *i* in this gradient as $h_i$, and this value normalized between zero (bottom of the gradient, area V1) and one. In summary:

$$J_s(i) = J_{min} + (J_{max} - J_{min})\ h_i \tag{11}$$

We assume therefore a gradient of values of $J_s$, with its value going from $J_{min}$ to $J_{max}$. Having large values of $J_s$ for association areas affects the spontaneous activity of these areas, even without considering inter-areal coupling. A good way to keep the spontaneous firing rate of these areas within physiologically realistic limits is to impose that the spontaneous activity fixed point is the same for all areas (*Murray et al., 2017b*). To introduce this into the model, we take into account that the solutions in the spontaneous state are symmetrical: $S_A = S_B = S$ (we assume zero noise for simplicity). The current entering any of the excitatory populations is then (assuming $I_{0A} = I_{0B} = I_0$):

$$I = (J_s + J_c)\ S + J_{EI} S_C + I_0 \tag{12}$$

Assuming a fast dynamics for $r_C$ and $S_C$ (mediated by GABA) as compared to $S_A$ and $S_B$ (mediated by NMDA) we can obtain the approximate expression for $S_C$:

$$S_C \simeq \tau_G\ \gamma_I\ r_C = 2 S J_{IE} \zeta + \beta \tag{13}$$

with

$$\zeta = \frac{\tau_G\ \gamma_I\ c_1}{g_I - J_{II}\ \tau_G\ \gamma_I\ c_1} \tag{14}$$

$$\beta = \tau_G\ \gamma_I\ \frac{c_1\ I_{0C} + g_I\ r_0 - c_0}{g_I - J_{II}\ \tau_G\ \gamma_I\ c_1} \tag{15}$$

The equation for the excitatory current has then the form

$$I = (J_s + J_c)\ S + 2\ J_{EI}\ J_{IE} \zeta\ S + J_{EI}\ \beta + I_0 \tag{16}$$

To maintain the excitatory input (and therefore the spontaneous activity level S) constant while varying $J_s$ across areas, we just have to keep the quantity $J_s + J_c + 2\ J_{EI}\ J_{IE}\ \zeta \equiv J_0$ constant (for the original parameters of the isolated area described above, we obtain $J_0 = 0.2112$ nA). A good choice, but not the only one, is to assume that the excitatory synapses to inhibitory neurons, $J_{IE}$, also scales with the ranks and with $J_s$ accordingly:

$$J_{IE} = \frac{1}{2\ J_{EI}\ \zeta}\ (J_0 - J_s - J_c) \tag{17}$$

This linear relationship ensures that the spontaneous solution is the same for all areas in the network. Note that deviations from this linear relationship would simply lead to different areas having slightly different spontaneous activity levels, but it does not substantially affect our main results.

Since $J_{IE}$ needs to be non-negative, the linear relationship above imposes a minimum value of $J_{min}$ = 0.205 nA for $J_s$. The particular maximum value of $J_s$, namely $J_{max}$, will determine the type of WM model we assume. Since the bifurcation point of an isolated area is at $J_s$ = 0.4655 nA for this set of parameter values, setting $J_{max}$ below that value implies that all areas in the network are monostable in isolation. In this situation, any sustained activity displayed by the model will be a consequence of a global, cooperative effect due to inter-areal interactions. On the other hand, having $J_{max}$ above the bifurcation point means that some areas will be multistable when isolated, for example they will be intrinsically multistable and compatible with classical WM theories.

Unless specified otherwise, we assume a range of $J_{min}$ = 0.21 nA and $J_{max}$ = 0.42 nA (i.e. below the critical value), so that the model displays distributed WM without having any intrinsically bistable areas.

## Computational model: inter-areal projections

We now consider the inter-areal projections connecting isolated areas to form the large-scale cortical network. Assuming that inter-areal projections stem only from excitatory neurons (as inhibitory projections tend to be local in real circuits) and that such projections are selective for excitatory neurons, the network or long-range input term arriving at each of the populations of a given area $x$ from all other cortical areas is given by

$$I_{A,\,net}^x = G \sum_y W^{xy} SLN^{xy} S_A^y \tag{18}$$

$$I_{B,\,net}^x = G \sum_y W^{xy} SLN^{xy} S_B^y \tag{19}$$

$$I_{C,\,net}^x = \frac{G}{Z} \sum_y W^{xy} \left(1 - SLN^{xy}\right) \left(S_A^y + S_B^y\right) \tag{20}$$

Here, $G$ is the global coupling strength, $Z$ is a balancing factor, and $W$ is the connectivity matrix (more details given below). In these equations, a superindex denotes the cortical area and a subindex the particular population within each area. The sum in all equations runs over all cortical areas of the network (N = 30). Excitatory populations A and B receive long-range inputs from equally selective units from other areas, while inhibitory populations receive inputs from both excitatory populations. Therefore, neurons in population A of a given area may be influenced by A-selective neurons of other areas directly, and by B-selective neurons of other areas indirectly, via local interneurons.

$G$ is the global coupling strength, which controls the overall long-range projection strength in the network ($G$ = 0.48 unless specified otherwise). $Z$ is a factor that takes into account the relative balance between long-range excitatory and inhibitory projections. Setting $Z$ = 1 means that both excitatory and inhibitory long-range projections are equally strong, but this does not guarantee that their effect is balanced in the target area, due to the effect of local connections. Following previous work (*Murray et al., 2017b*), we choose to impose a balance condition that guarantees that, if populations A and B have the same activity level, their net effect on other areas will be zero –therefore highlighting the selectivity aspect of the circuits. Again, deviations from this balance condition do not strongly affect our results besides the appearance of small differences between populations A and B. Considering that the transfer function of inhibitory populations is linear and their approximately linear rate-conductance relationship, it can be shown that

$$Z = \frac{2c_1 \tau_G \gamma_I J_{EI}}{c_1 \tau_G \gamma_I J_{II} - g_I} \tag{21}$$

Aside from global scaling factors, the effect of long-range projections from population $y$ to population $x$ is influenced by two factors. The first one, $W^{xy}$, is the anatomical projection strength as revealed by tract-tracing data (*Markov et al., 2013*). We use the fraction of labelled neurons (FLN) from population $y$ to $x$ to constrain our projections values to anatomical data. We rescale these strengths to translate the broad range of FLN values (over five orders of magnitude) to a range more suitable for our firing rate models. We use a rescaling that maintains the proportions between projection strengths, and therefore the anatomical information, that reads

$$W^{xy} = k_1 \left( FLN^{xy} \right)^{k_2} \tag{22}$$

Here, the values of the rescaling are $k_1$ = 1.2 and $k_2$ = 0.3. The same qualitative behavior can be obtained from the model if other parameter values, or other rescaling functions, are used as long as the network is set into a standard working regime (i.e. signals propagate across areas, global synchronization is avoided, etc.) FLN values are also normalized so that $\sum_y FLN^{xy}$ = 1. While in-degree heterogeneity might impact network dynamics (*de Franciscis et al., 2011*; *Roxin, 2011*), this was done to have a better control of the heterogeneity levels of each area, and to minimize confounding factors such as the uncertainty on volume injections of tract tracing experiments and the influence of potential homeostatic mechanisms. In addition, and as done for the local connections, we introduce a gradient of long-range projection strengths using the spine count data: $W^{xy} \rightarrow \left( J_s \left( x \right) / J_{\max} \right) W^{xy}$, so that long-range projections display the same gradient as the local connectivity presented above.

The second factor that needs to be taken into account is the directionality of signal propagation across the hierarchy. Feedforward (FF) projections that are preferentially excitatory constitute a reasonable assumption which facilitate signal transmission from sensory to higher areas. On the other hand, having feedback (FB) projections with a preferential inhibitory nature contributes to the emergence of realistic distributed WM patterns (*Figure 4*) (see also previous work *Markov et al., 2014b*; *Tsushima et al., 2006*). This feature can be introduced, in a gradual manner, by linking the different inter-areal projections with the SLN data, which provides a proxy for the FF/FB nature of a projection (SLN = 1 means purely FF, and SLN = 0 means purely FB). In the model, we assume a linear dependence with SNL for projections to excitatory populations and with (1-SLN) for projections to inhibitory populations, as shown above.

Following recent evidence of frontal networks having primarily strong excitatory loops (*Markowitz et al., 2015*), it is convenient to ensure that the SLN-driven modulation of FB projections between frontal areas is not too large, so that interactions between these areas are never strongly inhibitory. In practice, such constraint is only necessary for projections from frontal areas to 8 l and 8 m (which are part of the frontal eye fields) and has little effect on the behavior of our model otherwise. The introduction of this limitation has two minor consequences: (i) it allows area 8 l and 8 m to exhibit a higher level of sustained activity during distributed WM –as their hierarchical position and recurrent strength are not strong enough to sustain activity otherwise, as previously suggested in anatomical studies (*Markov et al., 2013*; *Markov et al., 2014a*) and (ii) it slightly shifts the transition point in cortical space (see *Figure 4—figure supplement 1*). Unless specified otherwise (and in *Figure 4*, where the limitation is not considered for a cleaner study of the effects of inhibitory feedback), we consider that the SLN-driven modulation of FB projections to 8 l and 8 m is never larger than 0.4.

Deviations from our general assumptions could occur in other areas –for example, a slightly stronger CIB value to primary somatosensory areas could prevent sustained activity in area 2, as the evidence of such activity is still controversial (*Lemus et al., 2010*; *Rossi-Pool et al., 2016*; *Zhou and Fuster, 1996*).

## Gating mechanism

To implement a simple gating mechanism which modulates areas receptive to a particular type of input, we assume that, when the gate of a given area is 'open', the strength of incoming synaptic projections effectively increases by a quantity $g_s$. This reflects, in a simplified way, existing gating mechanisms based on the activation of input-specific dendritic compartments, in which activation of a specific dendritic branch increases the effect of synaptic afferents targeting such dendritic branch (*Yang et al., 2016*). The effects of such gating mechanism are shown in *Figure 2—figure supplement 3*.

## Acknowledgements

We thank Rishidev Chaudhuri, John Murray and Jorge Jaramillo for their support during the development of this work, and Henry Kennedy for providing the connectivity dataset.

## Additional information

### Funding

| Funder | Grant reference number | Author |
| --- | --- | --- |
| National Institutes of Health | R01MH062349 | Xiao-Jing Wang |
| Office of Naval Research | N00014-17-1-2041 | Xiao-Jing Wang |
| Simons Foundation | | Xiao-Jing Wang |
| Human Brain Project SGA 3 | 945539 | Jorge F F Mejías |
| National Science Foundation | 2015276 | Xiao-Jing Wang |

The funders had no role in study design, data collection and interpretation, or the decision to submit the work for publication.

### Author contributions

Jorge F Mejías, Conceptualization, Formal analysis, Funding acquisition, Investigation, Software, Validation, Writing – original draft, Writing – review and editing; Xiao-Jing Wang, Conceptualization, Formal analysis, Funding acquisition, Investigation, Validation, Writing – original draft, Writing – review and editing

### Author ORCIDs

Jorge F Mejías ⓘ http://orcid.org/0000-0002-8096-4891
Xiao-Jing Wang ⓘ http://orcid.org/0000-0003-3124-8474

### Decision letter and Author response

Decision letter https://doi.org/10.7554/eLife.72136.sa1
Author response https://doi.org/10.7554/eLife.72136.sa2

## Additional files

### Supplementary files
• Transparent reporting form

### Data availability

The current manuscript is a computational study, so no data have been generated for this manuscript. Modelling code has been uploaded to ModelDB.

The following dataset was generated:

| Author(s) | Year | Dataset title | Dataset URL | Database and Identifier |
| --- | --- | --- | --- | --- |
| Mejias JF | 2022 | Distributed working memory in large-scale macaque brain model | http://modeldb.yale.edu/267295 | ModelDB, 267295 |

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

# Appendix 1

## Simplified computational model: two areas

To provide a deeper intuition of the nature of distributed WM and the model ingredients which are fundamental for the phenomenon, we describe here a simplified version of our model which is suitable for theoretical analysis. We will first introduce a version of the model with two interconnected excitatory nodes, each of them following a rate dynamics:

$$\tau \frac{dr_1}{dt} = -r_1 + \phi \left( J_s\, r_1 + J_c\, r_2 + I \right) \tag{23}$$

$$\tau \frac{dr_2}{dt} = -r_2 + \phi \left( J_c\, r_1 + J_s\, r_2 + I \right) \tag{24}$$

Here, $r_1$, $r_2$ are the firing rates of node 1 and 2, with display self-connections of strength $J_s$ and cross-connections of strength $J_c$. Both areas have a relaxation time constant $\tau$ and receive a background input current $I$. The transfer function is a threshold-linear function, that is $\phi\left(I\right) = I - \theta$ if $I > \theta$ and zero otherwise.

The equation for the fixed point of the dynamics can be easily written as a function of the average firing rate of both units, namely R, which leads to the equation

$$R = \left( J_s + J_c \right)\, R + I - \theta \tag{25}$$

The above is only true in the linear regime, while for the subthreshold regime we get the trivial solution $R = 0$. Solving *Eq. 25* leads to the following expression

$$R = \frac{\theta - I}{J_s + J_c - 1} \tag{26}$$

*Equation 26* permits a positive solution for R if both numerator and denominator are positive (or both negative, but the solution in this case is less interesting biologically). Interestingly, if the cross-connection strength is large enough, we can have a fixed point solution with nonzero R even if $J_s < 1$, which would correspond to the case in which nodes would be monostable if isolated from each other. Distributed sustained activity can emerge, therefore, even in simple cases of two coupled excitatory nodes.

## Simplified computational model: N areas

We present now the case in which we have a large number N of connected excitatory nodes. We consider a fully connected network for simplicity and N = 30 nodes for the simulations. The activity of the nodes is given by

$$\tau \frac{dr_i}{dt} = -r_i + \phi \left( J\, \eta_i\, r_i + \frac{1}{N} \sum_{\substack{j=1 \\ j \neq i}}^{N} G\, r_j + I \right) \tag{27}$$

Here, the term $\eta_i$ is a monotonically increasing linear function that is used to introduce a gradient of connectivity strength across the network, with minimum value $\eta_1 = 0.55$ and maximum value $\eta_N = 0.85$. Other parameters values chosen for simulations are: $\tau = 20\ ms$, $I = 4.81$, $J = 0.91$. For the transfer function, we choose a sigmoidal function (to demonstrate robustness of the phenomenon observed before for the threshold-linear):

$$\phi\left(I\right) = \frac{S_{max}}{1 + \exp\left[-S_{sat}\,\left(I - I_0\right)\right]} \tag{28}$$

Parameters chosen for simulations are $S_{max} = 60$, $S_{sat} = 0.1$, $I_0 = 30$. For these parameters, an isolated node is bistable only if $\eta \cong 0.88$, which is above our chosen $\eta_N$ value and implies that all nodes are monostable in isolation.

This model admits an approximate mean-field solution if we define the network-average firing rate as $R = \frac{1}{N} \sum\limits_{i=1}^{N} r_i$ . Averaging *Equation 27* over units and using standard mean-field approximations like $\langle f(x) \rangle \approx f(\langle x \rangle)$, we arrive at

$$\tau \frac{dR}{dt} = -R + \phi \left( J \langle \eta\, r \rangle + GR + I \right) \tag{29}$$

To estimate the average of the product $\eta\, r$ over units, we can follow to approaches. First, we can assume independence between these two variables and accept that $\langle \eta\, r \rangle \approx \langle \eta \rangle \langle r \rangle = \eta_0 R$, with $\eta_0$ being the mean value of $\eta$. We will refer to this as the first-order mean field solution in the text, and is given by the following equations:

$$\tau \frac{dR}{dt} = -R + \phi \left( \left( J \eta_0 + G \right) R + I \right) \tag{30}$$

$$\tau \frac{dr_i}{dt} = -r_i + \phi \left( J \eta_i r_i + G R + I \right) \tag{31}$$

It is important to notice that *Eq. 30* is the real self-consistent mean-field solution, which can be solved numerically to find the fixed point solutions of our system. *Eq. 31*, on the other hand, is useful since it allows to obtain the fixed point solutions for any node 'i' that is connected to the network, and allows to explore the effect of the local heterogeneity $\eta$.

As an alternative to this solution, we can consider additional term in the dependence between the heterogeneity and firing rate of the nodes. For example, we can assume that the firing rate of units will grow proportionally to both $\eta$ and G, which is a plausible approximation that takes into account both the heterogeneity and the global interactions. We can write this as

$$r(\eta) \cong \alpha\, r_0 + \beta\, G\, \eta \tag{32}$$

This leads to the following expression:

$$\langle \eta\, r \rangle \approx \alpha\, r_0\, \eta_0 + \beta\, G\, \left\langle \eta^2 \right\rangle = \alpha\, r_0\, \eta_0 + \beta\, G\, \eta_1 \eta_N \tag{33}$$

The mean-field solution obtained, which we denote here as second-order solution, is given by

$$\tau \frac{dR}{dt} = -R + \phi \left( \left( J \alpha \eta_0 + G \right) R + J \beta\, G\, \eta_1\, \eta_N + I \right) \tag{34}$$

$$\tau \frac{dr_i}{dt} = -r_i + \phi \left( J \eta_i r_i + G R + I \right) \tag{35}$$

A choice of $\alpha = 0.94$ and $\beta = 10$ (which fulfills the recommendations for a perturbative approach, since $\alpha \approx 1$ and $\beta \eta_1 \eta_N \ll 1$) provides good results as *Figure 3* shows. However, both the first and second order solutions predict the emergence of distributed WM, and the choice of one or the other solution has only minor implications.

## Simplified model for activity-silent memory traces

Similar to the simplified model above, we consider a network of N = 30 nodes whose dynamics is described by

$$\tau \frac{dr_i}{dt} = -r_i + \phi \left( J u_i \eta_i r_i + \frac{1}{N} \sum\limits_{\substack{j=1 \\ j \neq i}}^{N} G u_j r_j + I \right) \tag{36}$$

Both local and long-range projections are now modulated by a variable, $u_i$, accounting for short-term synaptic plasticity (STF), described by

$$\frac{du_i}{dt} = \frac{P - u_i}{\tau_{fac}} + P \left( 1 - u_i \right) r_i \tag{37}$$

Transfer function and other parameters as above, unless specifically mentioned (see *Figure 3— figure supplement 1*).

# Appendix 2

## Data analysis: Overview

We developed a numerical method to estimate the number of stable distributed WM attractors for a particular set of parameters values of our model. This method, which follows simplified density-based clustering principles, is used to obtain the results shown in *Figures 5 and 6*. To allow for a cleaner estimation, we do not consider noise in neural dynamics during these simulations.

Our large-scale cortical network has 30 areas, with each of them having two selective excitatory populations A and B. Simply assuming that each of the areas can reach one of three possible states (sustained activity in A, sustained activity in B, or spontaneous activity) means that our model can potentially display up to three to the power of 30 attractor combinations. This number can be even larger if we refine the firing rate reached by each area rather than simply its sustained/non-sustained activity status. Since it is not possible to fully explore this extremely large number of possible attractors, we devised a strategy based on the exploration of a sample of the input space of the model. The core idea is to stimulate the model with a certain input pattern (targeting randomized areas) and registering the fixed point that the dynamics of the model converges to. By repeating this process with a large number of input combinations and later counting the number of different attractors from the obtained pool of fixed points, we can obtain an estimate of the number of attractors for a particular set of parameter values.

## Data analysis: Stimulation protocol

A given input pattern is defined as a current pulse of fixed strength ($I_{pulse}$ = 0.2) and duration ($T_{pulse}$ = 1 sec) which reaches a certain number P of cortical areas. Only one population (A or B, randomized) in each area receives the input, and the P cortical areas receiving the input are randomly selected across the top 16 areas of the spine count gradient. This decreases the amount of potential input combinations we have to deal with by acknowledging that areas with stronger recurrent connections (such as 9/46d) are more likely to be involved in distributed WM patterns than those with weaker connections (such as MT). P can take any value between one and $P_{max}$ = 16, and we run a certain number of trials (see below) for each of them. Different values of $I_{pulse}$ and $T_{pulse}$, as well as setting the randomly selected areas at a high rate initial condition instead of providing an external input, have been also explored and lead to qualitatively similar results. Similar approaches regarding the use of random perturbations in high dimensional systems have been successfully used in the past (*Sussillo and Barak, 2013*).

It is also important to consider that not all values of P have the same number of input combinations. For example, *P* = 1 allows for 16*2 = 32 different input combinations (if we discriminate between populations A and B), while *P* = 2 allows for 16*(16-1)*2 = 480 input combinations, and so on. For a given value of P, the number of possible input combinations $N_c$ is given by

$$N_c = 2^P \begin{pmatrix} P_{max} \\ P \end{pmatrix} = 2^P \frac{P_{max}!}{(P_{max}-P)!\,P!} \tag{38}$$

By summing all values of $N_c$ for *P* = 1, ...$P_{max}$, we obtain around 43 million input combinations, which are still too many trials to simulate for a single model configuration. To simplify this further, we consider a scaling factor $F_c$ on top of $N_c$ to bring down these numbers to reasonable levels for simulations. We use $F_c$ = 0.0002 (or 0.02% of all possible combinations) for our calculations, which brings down the total number of simulated input combinations to around 9000. Other options, such as decreasing $P_{max}$ and using a larger scaling factor ($P_{max}$ = 12, $F_c$ = 0.01 or 1% or all possible combinations) give also good results. Since the rescaling can have a strong impact for small P (yielding a number of trials smaller than one), we ensure at least one trial for these cases.

To guarantee the stability of the fixed points obtained during these simulations, we simulate the system during a time window of 30 s (which is much larger than any other time scale in the system), and check that the firing rates have not fluctuated during the last 10 s before we register the final state of the system as a fixed point.

## Data analysis: Estimating the number of attractors

The final step is to count how many different attractors have been reached by the system, by analyzing the pool of fixed points obtained from simulations. A simple way to do this is to consider that, for any fixed point, the state of each area can be classified as sustained activity in population A (i.e. mean firing rate above a certain threshold of 10 spikes/s), sustained activity in population B, or spontaneous activity (both A and B are below 10 spikes/s). This turns each fixed point into a vector of 30 discrete states, and the number of unique vectors among the pool of fixed points can be quickly obtained using standard routines in Matlab (e.g. 'unique' function).

A more refined way to count the number of attractors, which we use in this work, is to define the Euclidean distance to discriminate between an attractor candidate and any previously identified attractors. Once the first attractor (i.e. the first fixed point analyzed) is identified, we test whether the next fixed point is the same than the first one by computing the Euclidean distance $E_d$ between them:

$$E_d = \frac{1}{n} \sum_{i=1}^{n} \left( r_i^{new} - r_i^{old} \right)^2 \tag{39}$$

where n = 30 is the total number of areas in the network (only one of the populations, A or B, needs to be considered here). If $E_d$ is larger than a certain threshold distance ε, we consider it a new attractor. We choose $\varepsilon = 0.01$, which grossly means that two fixed points are considered as different attractors if, for example, the activity of one of their cortical areas differs by 0.5 spikes/s and the activity on all other areas is the same for both. The particular value of ε does not have a strong impact on the results (aside from the fact that smaller values of ε gives us more resolution to find attractors). When several attractors are identified, each new candidate is compared to all of them using the same method.

Both the first and the second method to count attractors deliver qualitatively similar results (in terms of the dependence of the number of attractors with model parameters), although as expected the second method yields larger numbers due to its higher discriminability.

