## [Editor Report]

The final revision of the manuscript addressed the remaining issues raised by the reviewers. They felt that the paper is an important contribution to the field, providing new and testable insights into the interaction between cortical areas during the memory delay and that the work is likely to become "an influential reference for future modeling efforts" and deserves publication in *eLife*.

---

## [Decision Letter]

**Decision letter after peer review:**

Thank you for submitting your article "Mechanisms of distributed working memory in a large-scale network of macaque neocortex" for consideration by *eLife*. Your article has been reviewed by 2 peer reviewers, and the evaluation has been overseen by a Reviewing Editor and Tirin Moore as the Senior Editor. The reviewers have opted to remain anonymous.

The manuscript was well received by the two reviewers who felt that the large-scale distributed model of working memory has major strengths and fills an important gap in the literature. However, they also had a number of reservations and made suggestions which must be addressed for the manuscript to be accepted for publication in *eLife*.

Both reviewers were concerned about the lack of consideration of recent work documenting the existence silent delay activity. The concern is that the proposed model relies heavily and exclusively on persistent attractor states and the impression the manuscript created that these states are the only current thinking about working memory.

The reservations raised by the two reviewers are summarized below and along with the original critiques will provide the guide to the revision, should you decide to revise the manuscript.

1. Both reviewers recommend that the work is re-framed by taking into account newer studies and asked the authors "consider changing the tone of manuscript, so that it doesn't come across as if persistent attractors are state-of-the-art thinking about working memory".

2. Abstract, Introduction and Discussion

Please clarify in the Introduction and in Discussion that you are testing one model of working memory and acknowledge that there are other models that consider more complex dynamics of activity recorded during the delay. Also, please incorporate into the Abstract and Introduction the effects of deactivation and resistance to the distractors tested by your model.

3. Please address the point questioning the idea that "anatomical constraints" actually play a critical role in the model. If they are indeed critical to the model, provide documentation.

4. Consider moving the simplified model into the Supplementary Materials.

5. Discuss whether and how the proposed model explains extensive reports of silent periods in delay activity. A related issue is to what extent the proposed model depends on persistent activity and whether it can incorporate an STP.

6. In the Discussion please expand the point already made in the manuscript that "silent activity periods associated with silent WM (Masse et al., 2019; Mongillo et al., 2008; Stokes, 2015) could also be due to distributed WM effects".

7. Please provide the definition of "persistent" activity and consider the recommendation to change "persistent" to "elevated" or to "delay activity. Please also address the comment that past observations of "persistent" activity were based on activity averaged across trials, rather than on a trial-by-trial basis.

*Reviewer #1 (Recommendations for the authors):*

Overall the interesting findings seem to be obfuscated by seemingly not so relevant ones. For instance, both the abstract and introduction seem to ignore a main finding, which is the deactivation of the attractor by inactivating the top area. CIB and more resilience to distractors is not properly introduced, either (however mentioned in the abstract). Instead, the authors give relevance to a concept already in the literature, ie bistability accomplished through inter-area connectivity (Eding et al., PNAS, Guo et al., Nature) and to the fact that the model is “anatomically constrained”, which it is not clear that is indeed the case.

Anatomical constraints.

The abstract reads: “we developed an anatomically constrained model” but it is not clear in what ways the anatomical data constrains the main model. Indeed, in a supplementary figure and in the simplified model the authors show that it does not seem to matter much “Similar conclusions can be obtained when the anatomical structure of the cortical network is changed -for example, by randomly shuffling individual projection strength values”. This raises the question of which of the new insights depend on this and other “biological constraints”. Namely: “counterstream inhibitory bias”, superiority of distributed WM in resisting distractors, deactivation of global attractor by silencing a top layer, inactivation relationship with specific areas, etc. Each finding should be accompanied by how they are affected by including or not specific “biological constraints”. If some, like anatomical connectivity, are not critical, then they obfuscates the main findings and worsens the overall readability of the paper (which is very good, but a bit long) and could be removed or moved to a supplementary figure showing unequivocally in what ways it does (not) constrain the model. It is somewhat acknowledged in the paper that the main driver of the findings is the gradient of recurrent excitation (“As a matter of fact, the relevant parameter here is the strength of synaptic excitation that varies across cortical space, in the form of a macroscopic gradient”), but this is not very clear at times. Again, if this is indeed the case, less emphasis should be given to the anatomical “constraints” and instead the relevant feature should be spelled out clearly and early on (in the abstract and intro)

Simplified model.

It is not very clear what we gain with this model, if not to show that with homogeneous coupling (instead of heterogeneous from experimental data, see above) similar findings are achieved. The model is motivated by “The above model, albeit a simplification of real brain circuits, includes several biologically realistic features, which makes it difficult to identify essential ingredients for the emergence of distributed WM.” This seems a good reason to remove the “biological constraints” from the “full model” (see above). Additionally, because of where this model is introduced in the paper, it becomes unclear when the simplified or the full model is used in the following figures. This could be improved if the simplified model was introduced only in the supplementary material or in a subpanel with a clear title. At a minimum, all captions should say if the full/simplified model were used.

Previous experimental literature.

Overall we feel that several studies were not properly considered. For instance, Guo et al., Nature is not cited properly, nor discussed. Note for example that also in this paper there was a model – in addition to clear empirical evidence – with different areas and similar concepts as the ones that are explored here.

Likewise, both "Cortical information flow during flexible sensorimotor decisions" Markus Siegel et al., Nature and also Panichello and Buschman, Nature were not considered in this study. In both studies, they recorded from several areas across the hierarchy (from visual cortex to PFC) during WM and DM, so they seem to be extremely relevant, especially to constrain the model further in future studies. For example, Panichello and Buschman show clear WM codes in V4, not present in the current model. Another example: "We observed (…) a sharp binary jump of activity, areas like LIP exhibited a more gradual ramping activity, resembling temporal accumulation of information in decision-making(Shadlen and Newsome, 2001)". Siegel et al., Science show very convincing evidence that this is actually not the case and the model does not seem to match the latencies reported here.

Of course, this mismatch between data and model is not very important and it does not reduce the value of the current model, but the authors should consider toning down claims like "strong agreement with" or "an excellent agreement with a large body of data, from decades of monkey neurophysiological" which occur throughout the study. The model is a great proof of concept that provides several important insights, but it is far from being in "strong agreement" with what happens in the brain.

Somatosensory WM.

Relatedly, the authors perform an experiment that simulates "somatosensory WM". While the question as to which areas trigger the global attractor is interesting and would deserve to be explored further, the way this is framed (i.e. studying different WM modalities) is misleading and should be adapted. Figure2—figure supplement2 shows that the same global attractor is engaged irrespectively of which area is stimulated. The evidence points otherwise (see Christophel et al., 2017). For example Figure2—figure supplement2 shows persistent activity in IT, which would not be expected for somatosensory WM?

Inactivations.

It would be nice to have a schematic of when this inactivation is performed (which we think it is throughout the trial), like in FIGURE 7. It seems that the point made in fig6 C needs the areas to be silenced in the opposite direction (ie hierarchical order) to be conclusive. Figure F seems important, as well as the result in G, but it is very confusing. We would consider simplifying it to show more clearly the relevant features/points made. Again: how much of the findings (in particular the "bowtie" analyses) here depend on the "anatomical constraints" is unclear.

"which is in agreement with classical prefrontal lesion studies(Curtis and D'esposito, 2004)" The cited paper does not do what the authors did in the model. This line should be removed of better explained

"In some cases, inactivating specific areas might even lead to a disinhibition of other areas and to a general reinforcement of the attractor". Again, unclear why this is. Does this depend on gradient of recurrent excitation, hierarchy location or anatomical connectivity?

Relationship with other mechanisms of working memory.

This paragraph, while important, seems a bit incomplete in the current form. In particular the part where activity-silent is discussed. The results presented here seem to depend strongly on the persistent activity hypothesis of working memory. Does it make sense to think about distributed attractors through short-term plasticity? The relationship with silent activity does not seem straightforward and this discussion failed to illuminate it.

In the next paragraph, the authors say "This also means that silent activity periods associated with silent WM (Masse et al., 2019; Mongillo et al., 2008; Stokes, 2015)could also be due to distributed WM effects. Optogenetic inactivations could be used to test this result." This is an interesting idea, but could be expanded a bit more. Intriguingly, the authors cite papers (Masse et al., 2019; Mongillo et al.,) of local circuits with actual activity-siment mechanism. Instead, the author should cite empirical evidence of silent periods, of which the model proposed here offers an alternative view. For example: Wolff et al., Nature Neuroscience (Human occipital cortex), Barbosa et al., Nature Neuroscience (monkey PFC) and Akrami et al., Nature, (rodent PPC) etc.

*Reviewer #2 (Recommendations for the authors):*

I am not suggesting that the authors overhaul their model and start over. But a re-write (and some changing of terms, see below) would serve them well. I would encourage the authors to consider changing the tone of manuscript so that it doesn't come across as if persistent attractors are state-of-the-art thinking about working memory. I suggest a more up-front acknowledging of the newer developments (as opposed to a single paragraph near the end of the Discussion) and that their work will focus on mechanisms that allow average activity to remain elevated. Right now, it reads as if "persistent activity" is everything, with a disclaimer near the end.

Finally, I encourage the authors to *not* use the term "persistent activity" (try elevated or sustained elevated activity or just "delay activity"). As noted above, there is evidence against persistent activity. But more to the point, there is little or no evidence *for* persistent activity. Virtually all of the work purporting such evidence averaged neural activity across multiple trials. Across-trial averaging masks more complex dynamics like gaps of no spiking. One cannot conclude persistent firing from averaged data. It can only be addressed in real time at the single trial level. Also, there is a no definition of "persistent". Is it a spike every 5 ms? Every 10 ms? Every 100 ms? Using a term like "persistent activity" when it is not well defined and for which there is little direct evidence muddies the waters and does not do a service to the field.

Other comments:

One cannot help but wonder how the hierarchical trends discussed here relate to other hierarchical trends. For example, there is a gradual progression from sensory-related activity to task-relevant activity as one ascends the hierarchy. Or the greater mixed selectivity in higher cortex. Maybe those are separate issues. But if the authors have any insights into how their model contributes to them, it would certainly add value to their manuscript.

Page 4: "LIP exhibited a more gradual ramping activity, resembling temporal accumulation of information in decision making (Shadlen and Newsome, 2001)". Again, this was state-of-the-art like a decade ago. It ignores more recent work by Pillow, Shenoy, and others showing that the ramp-up is not gradual. When examined on the single-trial level, the activity is instead a series of discrete state changes. This does not take anything away from the elegant and important work of Shadlen and Newsome, without which the newer work would not have been possible. But, again, by focusing on older, not newer, work, the authors are not giving a full account of where we are in 2021.

[Editors’ note: further revisions were suggested prior to acceptance, as described below.]

Thank you for resubmitting your work entitled "Mechanisms of distributed working memory in a large-scale network of macaque neocortex" for further consideration by *eLife*. Your revised article has been evaluated by Tirin Moore (Senior Editor) and a Reviewing Editor.

The manuscript has been improved but there are some remaining issues that need to be addressed, as outlined below:

Reviewers were largely satisfied with the revised manuscript with one exception. They were concerned about the discussion of the role of plasticity in Attractor and Activity-Silent models (lines 559-566). It was felt that the work referred to in the Wang recent review which showed more spiking activity during manipulation of working memory, did not rule out synaptic plasticity. Furthermore, it was pointed out that Activity-Silent models also predict that spiking may be used to "ping" the network and read out the memories. In this case, the role of plasticity is to HELP the spiking, not to replace it.

To address this concern, this section should be modified. One option would be to provide clear evidence that synaptic plasticity only holds for Activity-Silent models and is not required by the attractor models. This can be done by citing specific references with the appropriate simulations/data or by providing new simulations/data.

Alternatively, this paragraph should be modified by allowing short-term plasticity to play a role in both types of models.

*Reviewer #1 (Recommendations for the authors):*

With most of my initial concerns addressed and the inclusion of interesting, new simulations, I fully support the publication of the manuscript in the current form.

*Reviewer #2 (Recommendations for the authors):*

The authors' revisions are mostly adequate.

However, the statements that activity-silent models " (1) it cannot filter out distractors that occur later in time than behavioral relevant stimuli, (2) it does not have a severely limited capacity (a characteristic of working memory) and (3) it is incapable of internal manipulation of information" is not true.

The activity-silent models can explain all of this. Synaptic weight changes are driven and refreshed by spiking. Thus, they have the same features and same control as attractor-state models. 1. Distractors can be filtered out by controlling spiking. 2. They do have a severe capacity limitation due to limitations in the spiking refresh rate. Multiple memories cannot be in the active state at the same time. That leads to capacity limitations. 3. Manipulation of WM is achieved by controlling spiking episodes, just like the attractor-state models.

The issue is that in testing the activity-silent models, the author has shifted too much of the burden to synapses alone. That is a misrepresentation of the activity-silent models. It is easy to refute a model if one makes a straw model of it. In the activity-silent models, synapses don't do everything. The help activity by briefly carrying the memories between spiking. That is why they are also referred to as "synaptic attractor" models. Because they also involve attractor states, they have many of the same features and mechanisms as attractor-state models. As a wise colleague recently said, the attractor-state and synaptic-attractor models are more similar than different. The characterization that the former can explain a variety of WM phenomena but the latter cannot is not accurate.

I think this is a valuable review. It is well-written. Attractor dynamics are indeed important and the review offers important insights. But surely these insights can be offered with misrepresenting other models.

---

## [Author Response]

Reviewer #1 (Recommendations for the authors):Overall the interesting findings seem to be obfuscated by seemingly not so relevant ones. For instance, both the abstract and introduction seem to ignore a main finding, which is the deactivation of the attractor by inactivating the top area. CIB and more resilience to distractors is not properly introduced, either (however mentioned in the abstract). Instead, the authors give relevance to a concept already in the literature, ie bistability accomplished through inter-area connectivity (Eding et al., PNAS, Guo et al., Nature) and to the fact that the model is "anatomically constrained", which it is not clear that is indeed the case.

We acknowledge that some important findings were not properly highlighted in our previous version. We have now highlighted more explicitly, in the introduction, our findings regarding CIB, distractor resilience, and the control/inactivation of distributed attractors by prefrontal areas (also in the abstract now, together with the other two).

We consider, however, that the focus on the distributed bistability via large-scale cortical networks is well placed, given that while the concept was already present in the literature (e.g. Christophel et al., 2017, Leavitt et al., 2017, Guo et al., 2017) our model is the first one to provide a mechanistic and anatomically-constrained explanation about the wide multi-area distribution reported in Christophel et al., 2017 and Leavitt et al., 2017. This goes beyond relatively simpler two-area interactions (Edin et al., 2009, Guo et al., 2017, Murray et al., 2017), which can’t explain the wide activity distribution and lack the spatial extension to predict global trends like the robust transient or the counterstream inhibitory bias.

We have also elaborated on the anatomical constraints of our model and discussed previous related work (Edin et al., 2009, Guo et al., 2017), as discussed below.

Anatomical constraints.The abstract reads: "we developed an anatomically constrained model" but it is not clear in what ways the anatomical data constrains the main model. Indeed, in a supplementary figure and in the simplified model the authors show that it does not seem to matter much "Similar conclusions can be obtained when the anatomical structure of the cortical network is changed -for example, by randomly shuffling individual projection strength values". This raises the question of which of the new insights depend on this and other "biological constraints". Namely: "counterstream inhibitory bias", superiority of distributed WM in resisting distractors, deactivation of global attractor by silencing a top layer, inactivation relationship with specific areas, etc. Each finding should be accompanied by how they are affected by including or not specific "biological constraints". If some, like anatomical connectivity, are not critical, then they obfuscates the main findings and worsens the overall readability of the paper (which is very good, but a bit long) and could be removed or moved to a supplementary figure showing unequivocally in what ways it does (not) constrain the model. It is somewhat acknowledged in the paper that the main driver of the findings is the gradient of recurrent excitation ("As a matter of fact, the relevant parameter here is the strength of synaptic excitation that varies across cortical space, in the form of a macroscopic gradient"), but this is not very clear at times. Again, if this is indeed the case, less emphasis should be given to the anatomical "constraints" and instead the relevant feature should be spelled out clearly and early on (in the abstract and intro)

It is important to clarify the differences between anatomically-constrained model and anatomically-constrained result. Our model is clearly anatomically constrained, in the sense that we use anatomical data to determine many of its parameters, such as the connectivity matrix or some local properties. However, having an anatomically-constrained model doesn’t necessarily imply that all results from such a model will depend critically on the particular anatomical values used. Indeed, some flexibility in parameter values should be expected to allow, for instance, inter-subject variability, or even robustness of the results across species. In general, results which are sensitive to changes in structural assumptions might be valid predictions for macaques, while results more resilient to these changes (like the emergence of distributed WM patterns, as Figure 3 shows) are more likely to occur in other species too.

Likewise, anatomical constrains will be more important for some brain areas but less for others. For example, the local synaptic strength assumed for V1 is not very important, as long as it remains at the bottom of the hierarchy.

In the supplementary figure mentioned by the reviewer (Figure 4 supplement 2e-h), we indeed observe that we obtain the same result *globally* (i.e. emergence of distributed WM patterns) when individual projection strengths are shuffled. However we also observe a clear effect in results *for individual areas* (see panel ‘f’, for example). More precisely, we see, as also stated in the text, that the duration of persistent activity for some areas is affected; other areas stop participating in the distributed attractor. Therefore, the anatomical constraint of projection strengths has a quantitative effect –areas may drop from the distributed WM pattern if their connectivity is altered –and therefore this constrain cannot be dropped without altering the predictions of the model.

A similar point can be made about the results of the simplified model: they indicate that the precise anatomical connectivity is not necessary to have distributed activity patters in a generic network, but the particular connectivity matters if we consider differences between concrete areas. Without the anatomical connectivity, we would not be able to make any claims about how areas like LIP, 8B or 9/46d participate or not in the distributed WM attractors, as showed in Figures 2 or 6 for example.

Following the reviewer’s advice, but also with this consideration in mind, we have added a sentence in the first paragraph of the results to summarize our clarifications from above. We have also carefully revised all the results of the manuscript and indicated, in a new paragraph in the discussion (page 15), the dependency of the results with the anatomical constrains. In particular:

– The CIB results in Figure 4 have already been shown (in Figure 3) to be the result of the concrete shape of the local strength gradient (linear vs spine count), and also not strongly dependent on the connectivity values (as similar results are found with the simplified model).

– The emergence and properties of a large number of attractors (Figure 5) are strongly dependent on the gradients, connectivity and also other choices of the model (in particular, the consideration of only two selective populations per area in the model). Therefore, although we expect that a large number of attractors will be found using, for example, connectomes for other species, we can’t predict their numbers.

– For a similar reason, the particular effects of silencing areas on the activity and number of attractors (Figures 6 and 7) will depend on the gradient and connectivity properties. These results will be therefore valid for macaque brains, but may differ for other species in which, for example, a bowtie structure is not present.

– The resistance of distributed WM patterns to distractors in Figure 8 is a property inherently linked to the existence of distributed WM patters, moderate values of Jmax, and the CIB, so as long as these two conditions are present, models of other animals’ brains and/or conditions will also display a similar behavior.

Simplified model.It is not very clear what we gain with this model, if not to show that with homogeneous coupling (instead of heterogeneous from experimental data, see above) similar findings are achieved. The model is motivated by "The above model, albeit a simplification of real brain circuits, includes several biologically realistic features, which makes it difficult to identify essential ingredients for the emergence of distributed WM." This seems a good reason to remove the "biological constraints" from the "full model" (see above). Additionally, because of where this model is introduced in the paper, it becomes unclear when the simplified or the full model is used in the following figures. This could be improved if the simplified model was introduced only in the supplementary material or in a subpanel with a clear title. At a minimum, all captions should say if the full/simplified model were used.

The apologize if the benefits of the simplified model were not properly explained, but we do not think that is a good idea to “remove the biological constraints from the full model”. These are two different and complementary levels of description, and they both provide useful information to the reader.

In particular, the simplified model tells us about the basic ingredients to have distributed WM (in a generic network, with simplified dynamics, etc) as explicitly stated in page 5: strong enough long-range projections and linear gradient of local couplings (or a CIB, if the gradient is sublinear). Such synthesis is not easy to do from the full model, and it could be useful, for example, to generalize to other systems in future studies (for example, brains of rodents or humans). On the other hand, the full model is needed to assess the validity of our claims in more realistic scenarios. If we only consider the results of the simplified model, it would not be clear whether these results would hold when adding more realistic considerations (i.e. real connectivity matrix, inhibitory populations, etc).

A good example from the text: the simplified model shows that distributed WM emerges for linear gradient of local recurrent strengths (Figure 3b,c). When we introduce the data about dendritic spines (Figure 3e,f), we discover that the soft saturation present in this data turns the distributed WM pattern into an unrealistic all-or-none situation. This leads us to identify CIB (or weaker excitatory FB projections, in the case of the simplified model) as a solution to have realistic WM patters in the full model (as later explored in Figure 4). The study of the simplified model alone, without data to constrain the gradient, would have not explored these effects, and the study of the full model alone might have overlooked the critical importance of CIB.

We therefore consider that both the full model and the simplified model provide important information to the reader, and we have a strong preference to maintain Figure 3 in the main text. In addition, we have now introduced a variation of the simplified model (new Figure 3 —figure supplement 1) to explore the case of distributed activity-silent memory. See response to another related comment below.

We have improved our justification for the use of the simplified model (page 5). We have also indicated clearly in the main text (page 6) that, after Figure 3, the full model is used for the rest of the paper. Furthermore, we added ‘full model’ in captions of Figures 4 and 5 to further emphasize this.

Previous experimental literature.Overall we feel that several studies were not properly considered. For instance, Guo et al., Nature is not cited properly, nor discussed. Note for example that also in this paper there was a model – in addition to clear empirical evidence – with different areas and similar concepts as the ones that are explored here.Likewise, both "Cortical information flow during flexible sensorimotor decisions" Markus Siegel et al., Nature and also Panichello and Buschman, Nature were not considered in this study. In both studies, they recorded from several areas across the hierarchy (from visual cortex to PFC) during WM and DM, so they seem to be extremely relevant, especially to constrain the model further in future studies. For example, Panichello and Buschman show clear WM codes in V4, not present in the current model. Another example: “We observed (…) a sharp binary jump of activity, areas like LIP exhibited a more gradual ramping activity, resembling temporal accumulation of information in decision-making(Shadlen and Newsome, 2001)”. Siegel et al., Science show very convincing evidence that this is actually not the case and the model does not seem to match the latencies reported here.Of course, this mismatch between data and model is not very important and it does not reduce the value of the current model, but the authors should consider toning down claims like "strong agreement with" or "an excellent agreement with a large body of data, from decades of monkey neurophysiological" which occur throughout the study. The model is a great proof of concept that provides several important insights, but it is far from being in "strong agreement" with what happens in the brain.

Although we had previously cited Guo et al., multiple times in our manuscript, including an explicit mention to its importance to extend our model to thalamocortical loops in the future, we agree that this paper required more consideration. In addition to very relevant experimental evidence, they also presented a computational model, although consisting only of two interacting areas (similar in scope to Murray, Jaramillo and Wang 2017). In this sense, it provides a valuable starting point and we now cite it in the introduction, when other two-area WM models (Edin et al., 2009, Murray et al., 2017) are acknowledged. We also included it, together with Murray et al., 2017, in a new sentence in the discussion stating that these works previously explored inter-areal interactions to sustain WM in more limited (i.e. two-area) models.

In addition, we have now cited the studies of Siegel et al., and Panichello and Buschman. We agree with the reviewer about the importance of these works in our conclusions. Even though the tasks are different from the one we simulate with our model (and therefore their results are not directly comparable), we think that they should guide future improvements of the model once ingredients such attention and sensorimotor integration are carefully taken into account –and we have indicated this in the text (page 13). The sentence about LIP has been modified, also as a response to a comment from Reviewer 2. We have also toned down overly strong claims along the manuscript, including replacing mentions to ‘strong agreement’ with just ‘substantial agreement’ or simply ‘agreement’.

Somatosensory WM.Relatedly, the authors perform an experiment that simulates “somatosensory WM”. While the question as to which areas trigger the global attractor is interesting and would deserve to be explored further, the way this is framed (i.e. studying different WM modalities) is misleading and should be adapted. Figure2—figure supplement2 shows that the same global attractor is engaged irrespectively of which area is stimulated. The evidence points otherwise (see Christophel et al., 2017). For example Figure2—figure supplement2 shows persistent activity in IT, which would not be expected for somatosensory WM?

We agree, current experimental evidence suggests that stimulation of different areas (or via different modalities) shouldn’t necessarily lead to the same attractors, as it happens with the model. We believe that the reason for this “convergence to the same attractor” is the absence of a gating mechanism in the model, which would allow some areas to participate in modality-specific global attractors. This is already tested in an extension of our model, shown in Figure 2 supplement 4, which aims to provide an explanation for such differences. We have rewritten the text of this result (page 5) so that the need of additional considerations is now clearer to the readers. As a result of this change, Figure 2 supplementary figures 3 and 4 have been swapped.

Inactivations.It would be nice to have a schematic of when this inactivation is performed (which we think it is throughout the trial), like in Figure 7. It seems that the point made in fig6 C needs the areas to be silenced in the opposite direction (ie hierarchical order) to be conclusive. Figure F seems important, as well as the result in G, but it is very confusing. We would consider simplifying it to show more clearly the relevant features/points made. Again: how much of the findings (in particular the “bowtie” analyses) here depend on the “anatomical constraints” is unclear.

As the reviewer suspects, the inactivation is performed throughout the trial, we have now clearly stated in Figure 6 caption. Panel 6C shows indeed the effect of silencing the areas in the opposite hierarchical order, as the reviewer says, and the silencing is also trial-long: first a trial with the last area silenced, then a trial with last and second-to-last areas silenced, etc. The whole purpose with this approach is to test the resilience of attractors when targeting top hierarchical areas specifically for the inactivation. And as we already discussed in the point above, these results are strongly dependent on the anatomical data used –results won’t necessarily be the same for connectomes of rodents or humans, for example.

We have clarified the results of Figure 6C,F,G in the main text (pages 8-9) and in the figure caption. We have also improved the clarity of the figures and explanations of Fig6F and G (including the bowtie analysis and how it depends on the anatomical data).

“which is in agreement with classical prefrontal lesion studies(Curtis and D’esposito, 2004)” The cited paper does not do what the authors did in the model. This line should be removed of better explained

We have corrected this sentence.

“In some cases, inactivating specific areas might even lead to a disinhibition of other areas and to a general reinforcement of the attractor”. Again, unclear why this is. Does this depend on gradient of recurrent excitation, hierarchy location or anatomical connectivity?

The described effect has its origin in the hierarchical relationships between areas and the CIB –silencing a top area which is inhibiting lower ones might release the lower areas from the inhibition and increase their activity. We have included a new sentence to clarify it, in page 9.

Relationship with other mechanisms of working memory.This paragraph, while important, seems a bit incomplete in the current form. In particular the part where activity-silent is discussed. The results presented here seem to depend strongly on the persistent activity hypothesis of working memory. Does it make sense to think about distributed attractors through short-term plasticity? The relationship with silent activity does not seem straightforward and this discussion failed to illuminate it.In the next paragraph, the authors say “This also means that silent activity periods associated with silent WM (Masse et al., 2019; Mongillo et al., 2008; Stokes, 2015)could also be due to distributed WM effects. Optogenetic inactivations could be used to test this result.” This is an interesting idea, but could be expanded a bit more. Intriguingly, the authors cite papers (Masse et al., 2019; Mongillo et al.,) of local circuits with actual activity-siment mechanism. Instead, the author should cite empirical evidence of silent periods, of which the model proposed here offers an alternative view. For example: Wolff et al., Nature Neuroscience (Human occipital cortex), Barbosa et al., Nature Neuroscience (monkey PFC) and Akrami et al., Nature, (rodent PPC) etc.

In response to this comment, and also to the concerns of Reviewer #2, we have included new simulation results (new Figure 3 —figure supplement 1) showing how activity-silent memory traces could also emerge in a distributed fashion. In this case, the short-term synaptic efficacy is maintained during longer periods of time because of the long-range interactions between brain areas, rather than local recurrent input. We have rewritten parts of the text over the entire manuscript (and particularly in the introduction and discussion) to make more explicit the generality of our proposed mechanism for distributed WM beyond persistent activity, and also to improve the discussion regarding silent WM. We have also expanded our proposed optogenetic testing.

Regarding the optogenetic inactivations, we meant to suggest that they could be used to test whether information encoded in WM areas could survive short inactivations. We realize that the previous writing was confusing, and since we are addressing the activity-silent phenomenon elsewhere in the text, we have now removed this sentence.

Reviewer #2 (Recommendations for the authors):I am not suggesting that the authors overhaul their model and start over. But a re-write (and some changing of terms, see below) would serve them well. I would encourage the authors to consider changing the tone of manuscript so that it doesn't come across as if persistent attractors are state-of-the-art thinking about working memory. I suggest a more up-front acknowledging of the newer developments (as opposed to a single paragraph near the end of the Discussion) and that their work will focus on mechanisms that allow average activity to remain elevated. Right now, it reads as if "persistent activity" is everything, with a disclaimer near the end.

Following the reviewer’s advice, we have adapted the text along the manuscript (and especially in the introduction and discussion) to put our proposal in a more general light and acknowledging other WM mechanisms from the beginning. Our work reads now as primarily focused on the persistent activity hypothesis for practical matters but highlighting that our distributed WM proposal can be applied more generally.

To elaborate on this point a bit further, we have also investigated, using a variation of our simplified model, whether our distributed WM proposal could also facilitate the emergence of activity-silent memory traces. More precisely, we have reduced the overall synaptic strength in our simplified model and included short-term synaptic facilitation in both local and long-range connections, and tested whether activity-silent memory traces can also benefit from inter-areal interactions and ‘silent’ distributed attractors may emerge. This seems to be the case, as we show in new Figure 3 —figure supplement 1. Our model shows that silent memory traces emerge when brain areas are allowed to support each other, but it fades away if we only consider isolated areas, as in the case of the persistent activity model. While this result opens the door for more realistic simulations, we hope that this will suffice to make a point about the generality of the distributed WM hypothesis proposed here.

Finally, I encourage the authors to not use the term "persistent activity" (try elevated or sustained elevated activity or just "delay activity"). As noted above, there is evidence against persistent activity. But more to the point, there is little or no evidence for persistent activity. Virtually all of the work purporting such evidence averaged neural activity across multiple trials. Across-trial averaging masks more complex dynamics like gaps of no spiking. One cannot conclude persistent firing from averaged data. It can only be addressed in real time at the single trial level. Also, there is a no definition of "persistent". Is it a spike every 5 ms? Every 10 ms? Every 100 ms? Using a term like "persistent activity" when it is not well defined and for which there is little direct evidence muddies the waters and does not do a service to the field.

This is a very interesting point, although we think that the situation might be a bit different in our case. Our model considers the macroscopic activity of brain areas, which in a real brain would be obtained by averaging over the activity of many individual neural responses in the same circuit. While single-neuron persistent activity is limited as explanation for WM, a ‘population persistent activity’ as used in our model is more plausible, and it could also arise via more flexible mechanisms which allow for a dynamics and highly-variable single-neuron activity (Goldman, Neuron 2009).

In addition, we consider that persistent activity should not be understood as a constant, fixed value of all firing rates, but as the opposite to decaying transient activity. This has been recently discussed in Wang, TiNS 2021, and it could indeed constitute a working definition for the term ‘persistent activity’. In general, persistent activity can incorporate complex dynamics and variability in the firing rates during the delay epoch, a feature of persistent activity which has been addressed in previous studies. For example, Compte et al., 2000 (see panel A in Author response image 1) already showed the presence of rhythmic variability during the persistent activity period. Such rhythmic activity is similar to the periodic bursts required for silent-activity mechanisms –as a matter of fact, Mongillo et al., (2008) showed such an example (see panel B) with a slightly different model parameter change from that corresponding to the activity-silent state regime.

**Author response image 1. sa2fig1:** 

Nonetheless, we have followed the advice and adapted the terminology so that the terms ‘sustained activity’ or ‘sustained delay activity’ are used by default, and included a new section in the discussion (page 13) where these issues are explained.

Other comments:One cannot help but wonder how the hierarchical trends discussed here relate to other hierarchical trends. For example, there is a gradual progression from sensory-related activity to task-relevant activity as one ascends the hierarchy. Or the greater mixed selectivity in higher cortex. Maybe those are separate issues. But if the authors have any insights into how their model contributes to them, it would certainly add value to their manuscript.

These are indeed relevant issues, as our work establishes a partial connection between structural gradients (dendritic spine count, position in the SLN-defined anatomical hierarchy) and functional ones (persistent activity being more common in higher vs lower areas in the hierarchy). Although the insight provided by our work is limited, the previous version of our manuscript provided an attempt (in the discussion) to relate our work with other hierarchical trends. We have now extended such paragraph to include the example of sensory- and task-related gradients and the mixed selectivity examples that the reviewer mentioned (page 11).

Page 4: "LIP exhibited a more gradual ramping activity, resembling temporal accumulation of information in decision making (Shadlen and Newsome, 2001)". Again, this was state-of-the-art like a decade ago. It ignores more recent work by Pillow, Shenoy, and others showing that the ramp-up is not gradual. When examined on the single-trial level, the activity is instead a series of discrete state changes. This does not take anything away from the elegant and important work of Shadlen and Newsome, without which the newer work would not have been possible. But, again, by focusing on older, not newer, work, the authors are not giving a full account of where we are in 2021.

Following the point of population activity of our previous comment above, we think it’s important to clarify that our model focuses on population-level (rather than neuron-level) dynamics, and therefore our description is not invalidated by the recent work by Pillow, Shenoy and others. We have modified the sentence highlighted by the reviewer to make this parallelism clearer in the text (page 4).

[Editors’ note: what follows is the authors’ response to the second round of review.]

Reviewer #1 (Recommendations for the authors):With most of my initial concerns addressed and the inclusion of interesting, new simulations, I fully support the publication of the manuscript in the current form.

We thank Reviewer 1 for his work to improve our manuscript.

Reviewer #2 (Recommendations for the authors):The authors' revisions are mostly adequate.However, the statements that activity-silent models " (1) it cannot filter out distractors that occur later in time than behavioral relevant stimuli, (2) it does not have a severely limited capacity (a characteristic of working memory) and (3) it is incapable of internal manipulation of information" is not true.The activity-silent models can explain all of this. Synaptic weight changes are driven and refreshed by spiking. Thus, they have the same features and same control as attractor-state models. 1. Distractors can be filtered out by controlling spiking. 2. They do have a severe capacity limitation due to limitations in the spiking refresh rate. Multiple memories cannot be in the active state at the same time. That leads to capacity limitations. 3. Manipulation of WM is achieved by controlling spiking episodes, just like the attractor-state models.The issue is that in testing the activity-silent models, the author has shifted too much of the burden to synapses alone. That is a misrepresentation of the activity-silent models. It is easy to refute a model if one makes a straw model of it. In the activity-silent models, synapses don't do everything. The help activity by briefly carrying the memories between spiking. That is why they are also referred to as "synaptic attractor" models. Because they also involve attractor states, they have many of the same features and mechanisms as attractor-state models. As a wise colleague recently said, the attractor-state and synaptic-attractor models are more similar than different. The characterization that the former can explain a variety of WM phenomena but the latter cannot is not accurate.I think this is a valuable review. It is well-written. Attractor dynamics are indeed important and the review offers important insights. But surely these insights can be offered with misrepresenting other models.

We thank the Reviewer for elaborating on this point and help us to clarify the text. The Reviewer is correct in that all three limitations mentioned above disappear when a combination of short-term plasticity and spiking activity are considered.

Importantly, we did not mean to imply that self-sustained activity and short-term facilitation are incompatible. As a matter of fact, short-term facilitation is part of recurrent synaptic interactions that need to be sufficiently strong for maintaining self-sustained firing. The attractor scenario is differentiated from the activity-silent scenario not by the nature of the underlying biological feedback mechanism (e.g. NMDA receptor dependent transmission or synaptotagmin 7 dependent synaptic facilitation), but by whether it is above a threshold strength.

To clarify this point and avoid misrepresenting the activity-silent models, we have slightly modified the title of the corresponding subsection (to ‘Attractor model of working memory and activity-silent state models’), and replaced the sentences mentioned by the Reviewer (page 13) by the following:

“Another example is self-sustained repetition of brief bursts of spikes interspersed with long silent time epochs (Mi et al., 2017). [...] Short-term plasticity could therefore contribute to activity-silent memory traces but also to self-sustained activity.”